

# The Marinada Fall Wind in the Eastern Ebro Sub-basin: Physical Mechanisms and Role of the Sea, Orography and Irrigation

Tanguy Lunel[1], Maria A. Jimenez[2], Joan Cuxart[2], Daniel Martinez-Villagrasa[2], Aaron A. Boone[1], and Patrick Le Moigne[1]

[1]Centre National de Recherches Météorologiques, Université de Toulouse, Météo-France, CNRS, Toulouse, France
[2]Grup de Meteorologia, Universitat de les Illes Balears, Palma, Spain

**Correspondence:** Tanguy Lunel (tanguy.lunel@umr-cnrm.fr)

**Abstract.**

During the warm months of the year in Catalonia,the sea-breeze overcomes the coastal mountain range and the marine air reaches the eastern Ebro sub-basin. This phenomenon is called Marinada and has recently been thoroughly characterized for the first time by Jimenez et al. (2023), based on surface climatological data. However, the main physical mechanisms involved in its

arrival and propagation remain to be discovered. This study aims to understand how the Marinada is formed and how it interacts with the already developed atmospheric boundary layer. Surface and atmospheric observations are used in combination with the coupled surface-atmosphere model Meso-NH to reveal the mechanisms at play. It is shown that the Marinada is generated by the advection of cool marine air mass over the Catalan Pre-coastal Range by the action of the sea breeze and the upslope wind. This marine air mass then flows into the Ebro basin, creating what is known as the Marinada. The characteristics and

dynamics of the Marinada allow it to be classified as a fall wind. It is also shown that the arrival, propagation and decay of the Marinada is strongly dependent on the larger scale weather situation: westerlies or thermal low. The current study provides a consistent framework for understanding the Marinada, paving the way for better modelling and prediction of the phenomenon.

**Keywords.** Mesoscale Circulation, Sea Breeze, Surface–Atmosphere Interactions, Downslope Wind Dynamics, LIAISE, Meteorological Modeling, Atmospheric Boundary Layer,

## 1 Introduction

Regional wind patterns significantly influence many aspects of the environment in which societies live. In the face of growing environmental challenges, understanding these wind dynamics is crucial for a deeper comprehension of the complex systems of our planet. More than just a meteorological curiosity, regional winds affect essential aspects such as extreme weather events, agriculture, human thermal comfort, air quality and energy production. Unravelling the details of these local wind patterns

improves our ability to accurately model, predict and adapt to weather and climate change.

In terms of climate change, the Mediterranean region is expected to be a high-stake region, with significant changes in weather patterns and the hydrological cycle, resulting in high risks for humans and ecosystems (Ali and Cramer, 2022).



Mesoscale circulations in the Mediterranean region can be caused by a variety of conditions: synoptic weather situation, local or large-scale relief, interaction with the sea, etc., or by a combination of these factors.

As the Mediterranean region experiences dry and hot summers, sea breezes are particularly common in coastal areas during the warm months of the year. Sea breezes are due to the daytime differential heating between the land and the sea, resulting in a lower pressure over the heated land surface than over the sea, and finally in a pressure gradient force (PGF) that accelerates the winds towards the land (Segal and Arritt, 1992; Miller et al., 2003; Crosman and Horel, 2010; Cuxart et al., 2014). The sea breeze systems typically consist of a stable low tropospheric zone with subsidence over the sea, a convective internal boundary

layer over the land, and a sea breeze front ahead (Miller et al., 2003; Crosman and Horel, 2010). The penetration length of the sea breeze is typically about 15 to 30 km (Pokhrel and Lee, 2011; Crosman and Horel, 2010) and can reach up to 85 km as documented by Simpson et al. (1977). In addition to the change in wind direction that characterises the arrival of the sea breeze, the sea breeze often coincides with a drop in temperature and an increase in humidity (Simpson et al., 1977; Crosman and Horel, 2010). In the Barcelona region in Catalonia, the characteristics of the sea breezes were studied by Redaño et al.

(1991). The authors showed that the sea breeze typically starts between 05:00 and 06:00 UTC (07:00 to 08:00 LST) and reaches its maximum speed between 12:00 and 13:00 UTC. This maximum wind speed ranges from 8 to 12 m s$^{-1}$ for 80 % of the sea breezes. The wind direction during the day is also described. It starts from the southeast at the onset of the sea breeze and gradually turns to the southwest under the influence of the Coriolis force.

The Mediterranean region is bordered by several mountain ranges, and orographically induced circulations also play a role

in many meteorological phenomena in this region. At local and regional scales, slopes strongly influence the circulations, mainly through slope winds (Orville, 1964; Haiden, 2003; Serafin and Zardi, 2010; De Wekker and Kossmann, 2015). The main driving force for upslope (downslope) winds is the PGF due to surface heating (cooling) (Haiden, 2003; Serafin and Zardi, 2010).

Since they share the same driving mechanism, sea breezes and slope flows can also interact. The interaction between the two

depends on the steepness of the slope and the stability of the atmosphere (Porson et al., 2007). For eastern Spain in summer, a region with moderate slopes and strong shallow convection activity, Miao et al. (2003) showed with a study case that the upslope flow has the potential to enhance the inland wind. In particular, it increases the depth of the sea breeze and accelerates the landward wind. The authors also investigated the influence of soil parameters on the sea breeze and showed that a very arid soil with dry vegetation could help the sea breeze penetrate up to 75 km inland, while a moist and vegetated land type limits

the inland sea breeze extension to 20 km.

On a regional scale, differences in the properties of air masses are also drivers of the low tropospheric circulations. In particular, differences in temperature and moisture between two different air masses lead to differences in air density and later to pressure gradients at the interface between air masses. This density difference has been identified as a driver of several downslope winds, such as the Santa Ana wind in California (Hughes and Hall, 2010), the Bora wind in Croatia (Jurčec, 1981;

Smith, 1987; Grisogono and Belusic, 2008), or the Mistral wind in southern France (Pettré, 1981; Drobinski et al., 2017). When the density difference is the main driver, such downslope winds are called fall winds, as in the case of Santa Ana winds or weak to moderate Bora wind events (Smith, 1987; Hughes and Hall, 2010). Although it belongs to the family of gravity flows (Mahrt,



1982), fall wind is used in contrast to katabatic wind, a term mainly applied to gravity flows whose potential temperature deficit is due to surface radiative cooling (Poulos and Zhong, 2008). These fall winds have in common a strong temperature inversion cap over an almost homogeneous cold layer, and this inversion cap can be used to determine the height of the flow (Pettré, 1981; Smith, 1987; Grisogono and Belusic, 2008; Hughes and Hall, 2010). The inversion cap separates the flow pouring down the slope from the pre-existing atmospheric boundary layer (ABL), and the distinction between a lower-flowing air mass and the ABL ultimately allows hydraulic theory to be used to understand the phenomena studied (Durran, 2003; Yu et al., 2005; Grisogono and Belusic, 2008).

The Ebro basin is a region located in northeastern Spain, surrounded by mountains, and is characterized by a semi-arid climate. Due to the longitudinal shape of the basin and the presence of the Pyrenees to the north, the winds are channeled into the basin and two wind directions dominate: northwest and southeast (Jiménez et al., 2009). Along the axis of Zaragoza - Reus is the lowest part of the basin, which actually follows the Ebro River, and where the channelling effect of the winds is most pronounced. It is also the zone where the Cierzo blows the strongest (Masson and Bougeault, 1996; Jiménez et al., 2009; Ortega et al., 2022). The Segre sub-basin is in the northeastern part of the Ebro basin, and has some differences in the wind patterns encountered. There, the Cierzo is weaker and comes from the west-southwest. The influence of local slope flows is also more pronounced (Martínez et al., 2008; Cuxart et al., 2012). Recently, for the first time, a wind known locally as the Marinada has been thoroughly characterised by Jimenez et al. (2023). It is a cool and humid wind that blows in late summer afternoons in the Segre sub-basin. Its characterisation was carried out using near-surface observations within the Segre sub-basin. However, a lack of atmospheric observations has prevented a characterization of its vertical extent. In addition, the processes that lead to the Marinada are not yet fully understood.

The aim of this paper is to characterize the Marinada over its entire height and to investigate the dynamic processes behind it. New observations from the Land surface Interactions with the Atmosphere over the Iberian Semi-arid Environment (LIAISE) campaign (Boone et al., 2019) are used in combination with a mesoscale meteorological model to improve the understanding of the Marinada phenomenon.

Following this introduction, the next section describes the observational material, the two case studies selected and the coupled surface-atmosphere model used for the study. The third section focuses on the results obtained. In particular, the Marinada seen on the Ebro basin side is linked to what happens earlier on the coast. Then, four different stages of the Marinada are identified and detailed for the two case studies. Finally, the influence of irrigation on the Marinada is shown. The fourth section discusses the results and is followed by the conclusion.

## 2 Materials and methods

### 2.1 Observational data

To study the surface behavior of the Marinada, the network of Automatic Weather Stations (AWS) of the Catalan Meteorological Service (or Servei Meteorològic de Catalunya (SMC)) is used (Servei Meteorològic de Catalunya, 2011). The SMC AWS network is fairly dense, with stations typically 15 km apart. They include temperature and humidity measurements at



1.5 m a.g.l. (above ground level) and wind measurements at 10 or 2 m a.g.l., as some stations were installed primarily for agro-meteorological purposes. Observations are aggregated into 30-minutes data and made available on the SMC website. The extensive coverage of this network provides a good representation of the spatio-temporal behavior of the Marinada over the course of the day at the surface. The SMC AWS used in the study are mapped in figure A1, and listed in Table A1.

For the irrigated and dry zones in the Segre sub-basin, a wide range of observations is available for the last two weeks of July 2021, thanks to the special observation period of the international LIAISE campaign (Boone et al., 2019). The aim of this campaign is to study the impact of anthropization on surface-atmosphere interactions and the water cycle in semi-arid environments. The many observations collected during this campaign are available in the open access LIAISE database (AERIS, 2021). Although the Marinada study was not one of the main objectives of the LIAISE campaign, a substantial number

of measurements can be used to study it. Furthermore, the Marinada was a phenomenon that was present on most days of the special observation period of the campaign and needs to be well characterized and understood for a proper analysis of the LIAISE data. This work particularly uses data from mast, radiosondes and wind profiler at Els Plans (Fig. 2). The mast at Els Plans was 50 m high and was equipped with several temperature, humidity and wind sensors at different heights (Price, 2023a). To facilitate comparison with the SMC network, the 10 m observations are used here. Radiosondes with temperature, humidity

and wind sensors were launched hourly during the daytime along the Special Observation Period (14 to 30 July) (Price, 2023b), some days until late, allowing the Marinada study on these days. An ultra-high frequency (UHF) radar is also installed at Els Plans, and the wind profile can be obtained up to 2000 m a.g.l. (Lothon, 2022).

## 2.2   Study area

The Marinada has already been investigated for its behavior observed at the surface with the SMC stations and for its part

located in the eastern Ebro basin (Jimenez et al., 2023), i.e. the Segre sub-basin. To gain a better understanding of the origin of the Marinada, the study area considered here has been extended to include the coastal area. For the purposes of this study, this region is divided into several relatively homogeneous parts in terms of topographic features. Starting from the sea and moving inland, we find the sub-regions described in Table 1 and located in Fig. 2a. These zones allow to highlight different influences and processes on the atmosphere depending on the corresponding surface.

The area of interest for the Marinada study can be schematically divided into three zones: a coastal zone near Torredembarra ("Sea" and "Coast" in Fig. 2a), an agricultural plain to the east of Lleida in the Ebro basin ("Irrigated" and "Dry" in Fig. 2a), and the Catalan Pre-coastal Range (CPR), which separates the two previous zones and runs on a southwest-northeast axis ("Northwest slopes", "Conca de Barberà" and "Alt Camp") in Fig. 2a). The Conca de Barberà is an area at a lower altitude than the rest of the CPR, making it a privileged path for the wind. It is bordered to the northwest by the Serra del Tallat, and to the

southeast by the Serra de Miramar. The Serra del Tallat is a low mountain sub-range to the west of which the Ebro basin begins. The part of the Ebro basin studied here is called the Segre sub-basin, and corresponds to the areas designated as "Northwest slopes", 'Dry", and "Irrigated" in Table 1 and Fig. 2a.

To describe the evolution of the Marinada across these sub-regions, a transect going from the sea to the Segre sub-basin is also used. The chosen transect passes through 5 points of interest, from the sea to the Ebro basin: Torredembarra, Serra de





| Areas names | Relief | Main land uses | Distance from the coast [km] |
|---|---|---|---|
| Sea | Flat | Sea | -15 − 0 |
| Coast | South-facing slope of 1% | Mixed: urban, forest and crops | 0 − 13 |
| Alt camp | South-facing slope of 1% | Rain-fed crops | 10 − 30 |
| Conca de Barberà | Closed basin | Rain-fed crops | 22 − 41 |
| Northwest slopes | Northwest-facing slope of 3-4% | Rain-fed crops and forests | 39 - 59 |
| Dry | Northwest-facing slope of 1% | Rain-fed crops | 49 − 65 |
| Irrigated | West-facing slope < 1% | Irrigated crops | 53 − 83 |

**Table 1.** Table of the study sub-regions shown in Fig. 2 and their topographical characteristics.

Miramar, Serra del Tallat, Els Plans and La Cendrosa. These points of interest are also shown in Fig. 2, and are summarized in Table 2. These sites correspond to key locations for characterizing the Marinada. Specifically, La Cendrosa and Els Plans sites are reference sites in the irrigated and dry areas respectively, with 50 m mast and radiosoundings at each.

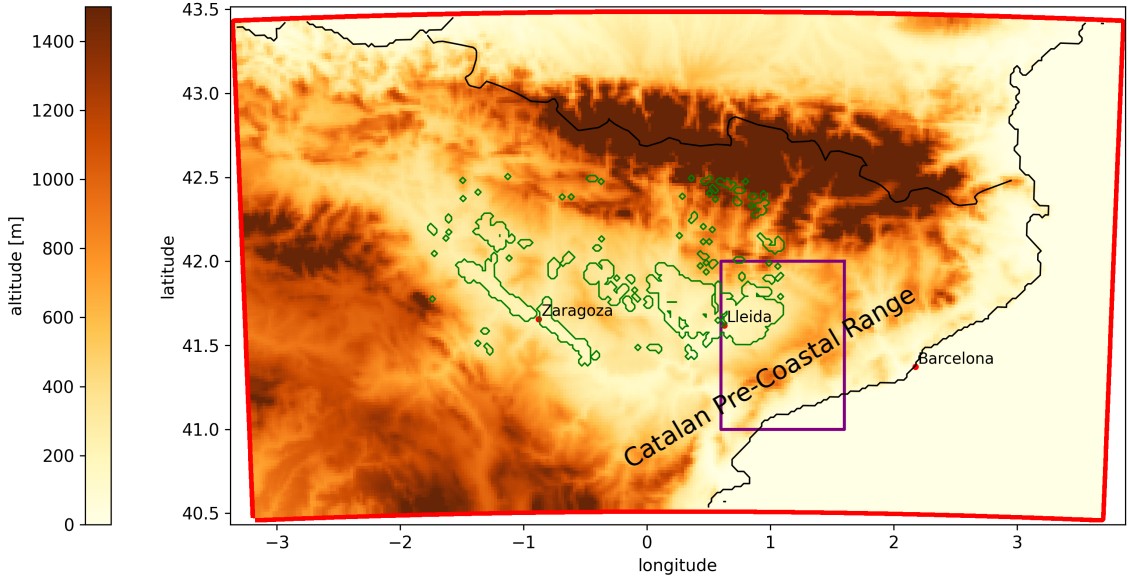

**Figure 1.** Domain of the model. The black lines represent the coastline or country borders, and the green lines delineate the areas of the Ebro basin irrigated. The purple rectangle delineates the study area for the Marinada.



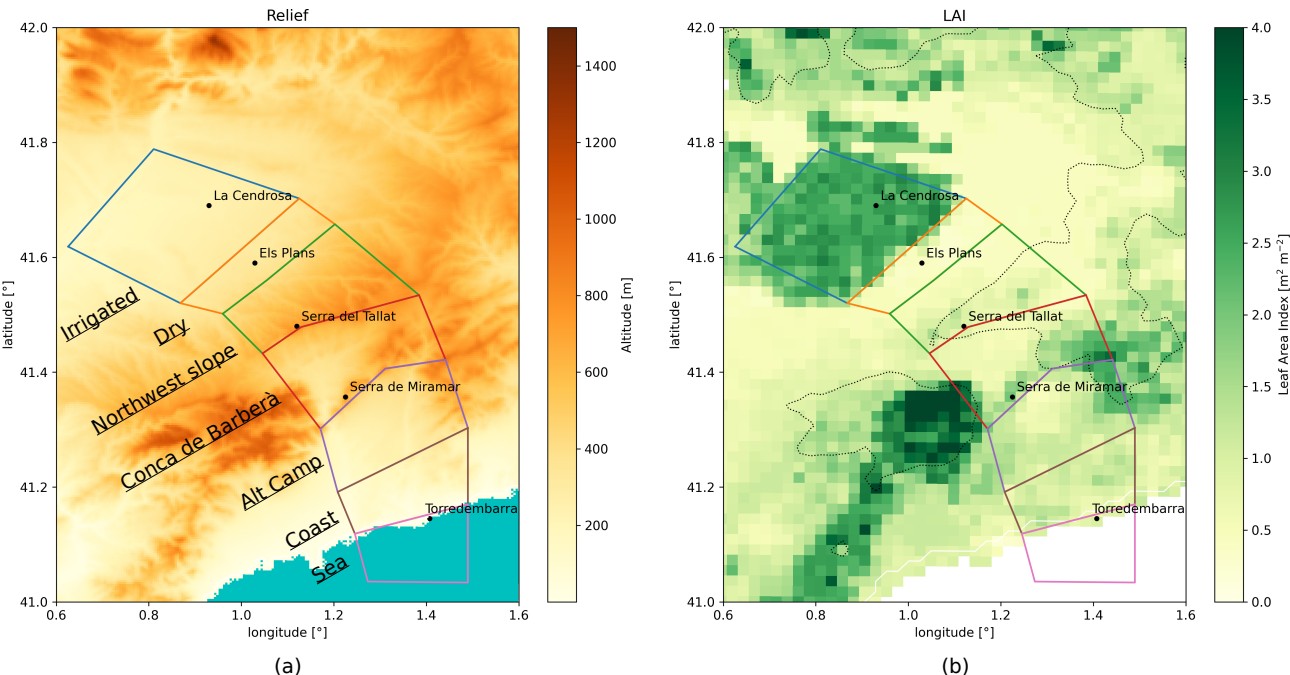

**Figure 2.** Topographic properties of the study area, corresponding to the purple rectangle of Fig. 1. Orography is shown in (a) and Leaf Area Index from the model in (b). The different areas described in Table 1 are shown as coloured polygons, with the area names underlined in (a). The dotted line in (b) follows the 600 m elevation isoline.

| Name | Acronym | Coordinates | Observation type | Topographic interest |
|---|---|---|---|---|
| Torredembarra | To. | 41.146°N, 1.418°E | SMC AWS | On the coast |
| Serra de Miramar | S.d.M. | 41.357°N, 1.225°E | No observation | Limit between Conca de Barberà and Alt camp |
| Serra del Tallat | S.d.T. | 41.480°N, 1.120°E | No observation | Eastern limit of Ebro basin |
| Els Plans | E.P. | 41.590°N, 1.029°E | LIAISE campaign data | At the foot of the CPR |
| La Cendrosa | L.C. | 41.693°N, 0.928°E | LIAISE campaign data | In the lower Segre sub-basin |

**Table 2.** Points of interest along the Torredembarra - La Cendrosa transect. The La Cendrosa and Els Plans sites had other in-situ instruments for studying the surface, which are not of interest for the Marinada study.



## 2.3 Weather situation of the two case study

In this case, two study days are considered: 16 and 21 July 2021. These two days correspond to two synoptic weather situations
common in summer in this part of Catalonia, and Marinada was present on both of these days, as shown in Jimenez et al. (2023).
The 16th represents a case of northwesterly flow in the Ebro valley. At 500 hPa, a ridge of high pressure stretches between
Spain and England, while a low pressure system settles over Switzerland. This creates a north-northwest wind at 500 hPa over
Catalonia. At the surface, the wind is channeled into the Ebro basin, giving it a westerly component in the lower layers over the
study region. This type of westerly wind situation is very common in the Ebro valley in summer (Capel Molina, 1999; Martínez
et al., 2008), and the Cierzo is a special case of these westerly winds (Ortega et al., 2022). The day of the 21st corresponds to
a thermal low over the Iberian Peninsula (Hoinka and De Castro, 2003). At 500 hPa, the action centers are quite far apart, and
a flow of about 10 m s$^{-1}$ blows in from the northwest. At 850 hPa, the wind blows from the southwest, parallel to the coast. In
the lower layers, the warming of the Ebro valley creates a thermal low, and thus a mean sea-level pressure gradient of -1.7 hPa
between Zaragoza and Reus, respectively in the central Ebro basin and on the coast, as described in Jimenez et al. (2023) and
Tudurí et al. (2003). This pressure gradient facilitates the existence of a southeasterly flow, which is then largely influenced by
other diurnal mesoscale circulations in the lower layers. This type of weather situation is rarer than westerly winds, but is also
common in summer in the region of interest (Favà et al., 2019). The present study details in depth the Marinada dynamics for
16 July, and then highlights the main differences with the 21 July.

## 2.4 Model set-up

### 2.4.1 Overview of the surface-atmosphere coupled model

To complement the above-mentioned observational data, a coupled surface-atmosphere limited-area model is used to delve into
the processes at play during Marinada formation. The atmospheric model is Meso-NH v5.5.1 (Lac et al., 2018) and is coupled
to the SURFEX v8.1 surface model (Masson et al., 2013). The model is run with a horizontal resolution of 2 km x 2 km and
covers the entire Ebro basin as well as the Pyrenees. The domain is shown in Fig. 1. Since irrigation is not represented in the
atmospheric analyses used for initialization, but it is added to the current model, a two-week spin-up period is run in order
to cool down and humidify the air in the Ebro basin. After the spin-up, from 14 to 22 July, the model runs are continuous
for each distinct irrigation parameterization detailed in Sect. 2.4.2. The timestep of the model is 3 s in order to respect the
Courant-Friedrichs-Lewy condition (Courant et al., 1956) between vertical layers in the convective updrafts.

The Meso-NH atmospheric model has 88 vertical layers from 0 m to 16,000 m a.g.l., with a tighter mesh near the ground
and a looser mesh at higher altitudes. The first level is at 2 m a.g.l. and the lowest 60 levels are below 2000 m, allowing fine
simulation of the ABL. The atmosphere is initialized every 6h and then forced to its boundaries by analyses from the European
Centre for Medium-range Weather Forecast (ECMWF) model. The turbulence is calculated using a 1D scheme (Cuxart et al.,
2000) and the mixing length of Bougeault and Lacarrère (1989). The Eddy-Diffusivity-Kain-Fritsch parameterization is used
for shallow convection (Pergaud et al., 2009). The radiation model employed in this study is a composite model that combines
the Morcrette (1991) scheme for shortwave fluxes and the Mlawer et al. (1997) scheme for longwave fluxes. This radiation



model is run at 5-minute intervals to ensure a gradual temporal evolution of incident radiation throughout the course of the day. The aerosol content within the model is calibrated to replicate the characteristics of a semi-arid region, with an optical aerosol thickness set at 0.2.

The surface model SURFEX is a collaborative software package that is currently being maintained and further developed

at CNRM (Centre National de Recherches Météorologiques) and LAERO (Laboratoire d'Aérologie), in partnership with other international collaborators from the ACCORD Consortium (A Consortium for COnvection-scale modelling Research and Development – http://www.umr-cnrm.fr/accord/) who contribute to its enhancements (Masson et al., 2013). SURFEX offers a variety of model options tailored to different surface types. The four main surface types in SURFEX are seas, lakes, urban areas and natural land. Each of them calls a different model. In this particular study, we employ straightforward parameterizations

to model the sea and the lakes. These parameterizations utilize prescribed sea surface temperatures sourced from the forcing files and a roughness length determined by the Charnock formula (Charnock, 1955). Urban areas, though relatively sparse in the region under consideration, are modeled using the Town Energy Balance (TEB) approach as described by Masson (2000). The predominant land cover type in the region of interest consists of natural land. This natural land is sub-classified into 12 plant functional types, allowing a separate representation for different land-surface types, from bare soil to forest. To model

these different functional types the Interaction Soil Biosphere Atmosphere (ISBA) model is used (Noilhan and Planton, 1989; Noilhan and Mahfouf, 1996). ISBA is configured with 14 vertical soil layers, the deepest being 12 m below the surface. Soil characteristics are derived using a pedo -ransfer function (Cosby et al., 1984), which relies on soil texture information obtained from the database of Nachtergaele et al. (2010). Water and heat transfer are then simulated using a diffusive approach, allowing for the direct application of Fourier and Darcy laws as described by Decharme et al. (2011). The land-use classifications are

derived from the Ecoclimap-II database (Faroux et al., 2013), which is based on satellite data spanning the period from 1999 to 2005. Leaf area index (LAI) and albedo values are also extracted from this database, providing typical values for the end of July. Surface roughness characteristics are similarly obtained from the land-use classes. Photosynthetic activity and its impact on transpiration are represented through the $ISBA - A - g_s$ model (Calvet et al., 1998).

### 2.4.2 Irrigation representation

In addition to these typical ISBA model parameters, ISBA is modified to represent irrigation. The changes made to the model are twofold.

The first concerns the modification of the land surface characteristics in the irrigated zone. In Ecoclimap-II, each grid point corresponding to an irrigated area is identified and its surface characteristics are modified. In the irrigated grid point, the percentage of irrigated fields is set to 90%, with the rest divided between rocks, grass, shrubs, trees and C3 crops. Ecoclimap-II

LAI values were found to be underestimated within the irrigated zone for the month of July when compared to direct satellite LAI data (not shown). To correct for this bias and make the surface feature more consistent with the satellite data, the LAI value is randomly set between 2 and 3 for each irrigated grid point. The randomness of the value makes it possible to maintain some heterogeneity in the zone. The LAI map is shown in Fig. 2.




The second change concerns soil moisture. The modification of the soil moisture is only applied to the areas identified as
irrigated in Ecoclimap-II. Here three different parameterizations are applied:

- *NOIRR:* The first parameterization does not add water artificially. Irrigation is in this case only represented by the
  increase of LAI in the irrigated area. Soil moisture values are initialized using the ECMWF analysis at a two-kilometer
  resolution. Since the soil moisture is in this case very low, close to the wilting point, the modeled evapotranspiration is
  also low, and the turbulent heat fluxes from the surface to the atmosphere are similar to those of a non-irrigated area.
  This parameterization will therefore be referred to as NOIRR.

- *THOLD_IRR:* The second parameterization for soil moisture represents the irrigation through a fixed amount of water
  added when the soil moisture reaches a given threshold. The threshold is based on the soil water index (SWI) weighted
  by the root fraction (WSWI) defined by equation 1 and 2. When the WSWI goes below 0.5, irrigation is triggered.

$$SWI_i = \frac{w_{g_i} - w_{g_{wilt}}}{w_{g_{fc}} - w_{g_{wilt}}} \tag{1}$$

$$WSWI = \sum_{i=1}^{14} SWI_i * RootFrac_i \tag{2}$$

where $w_{g_i}$ is the current volumetric soil moisture in layer $i$, $w_{g_{wilt}}$ is the volumetric soil moisture at wilting point, and
$w_{g_{fc}}$ is the volumetric soil moisture at field capacity, all in m$^3$ m$^{-3}$. $RootFrac_i$ is the proportion of the plant roots
present in layer $i$.

The water amount applied is 30 mm at each irrigation event. The soil moisture heterogeneity in the irrigated area on the
morning of the 16th is obtained with a 15 days spin-up.

- *FC_IRR:* The last parameterization maintains the soil moisture of each soil layer at field capacity throughout the simu-
  lation. This is achieved by hard-coding the soil moisture into the model, rather than adding water through a physically
  meaningful process. The validity of such a parameterization will be developed in the Sect. 4.

Outside of the irrigated area, soil moisture values are initialized using the SWI values from the ECMWF analysis. The soil
data from the ECMWF analysis have been shown to give good results when used with Meso-NH–SURFEX (Noual et al.,
2023). By the morning of 16 July, simulated soil moisture levels are close to the wilting point across the rain-fed areas. The
soil moisture maps resulting from each parameterization are shown in Fig. 3. The parameterization *FC_IRR* is used in the
sections 3.1 to 3.4, and the three parameterizations are compared and discussed in Sect. 3.5.

## 2.5 Hydraulic theory and Froude number

As stated in Sect. 1, the hydraulic theory can be used to analyze downslope circulations. In particular the Froude number will
be used in the following to identify and distinguish flow regimes.





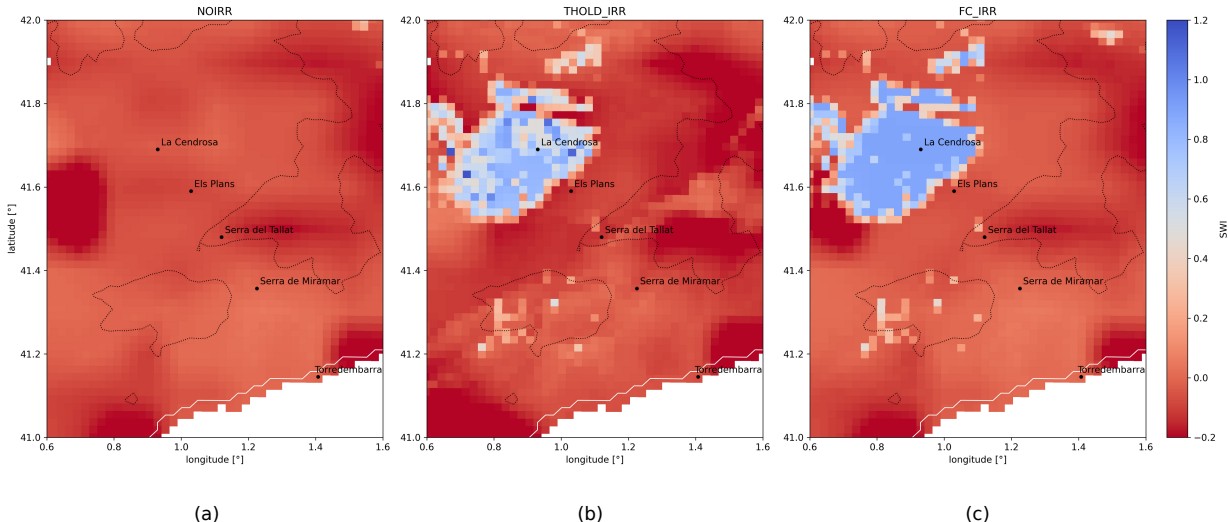

(a)                                      (b)                                      (c)

**Figure 3.** Maps of Soil Wetness Index (SWI) on the study area for the 3 different irrigation parameterizations on 16 July 2021, at 12:00 UTC. The dotted lines follow the 600 m elevation isoline.

The Froude number is the ratio of the flow kinetic energy to the external gravitational field. Depending on the field to which it is applied, different equations can be considered. In the present case, the definition used is that of Yu et al. (2005) and
Heinemann (1999) and is given in Eq. 3.

$$Fr_m = \frac{U_m{}^2 \, \theta_{top}}{g \, h_{top} \, (\theta_{top} - \theta_m)} \tag{3}$$

where $h_{top}$ is the height a.g.l. of the top of the jet, $U_m$ is the mean wind speed in the layer below $h_{top}$, $\theta_{top}$ is the potential temperature at $h_{top}$, $\theta_m$ is the mean potential temperature below $h_{top}$, $g$ is the gravitational acceleration.

To fit the present case, $h_{top}$ must be chosen to be above the temperature inversion cap of the downslope flow to account for
stability due to potential temperature stratification. It must also be above the jet nose, i.e. the maximum wind speed in the jet. Finally, the following definition is retained, which allows these two conditions to be met in most cases: $h_{top}$ is the height above the jet where the wind speed is equal to 70% of the jet's maximum wind speed. Using a lower percentage of the wind speed reduces the chances of finding an altitude that meets this definition, as the wind speed does not always decrease very much in the upper layer. On the other hand, using a higher percentage threshold does not allow the temperature inversion cap to be fully
captured.

The value of the Froude number can be directly interpreted in terms of the flow regime:



- *Fr > 1 – Supercritical regime*: the fluid is driven only by gravity, is faster than the wave speed, and is not affected by the downstream conditions.

- *Fr < 1 – Subcritical regime*: the fluid is driven by the downstream conditions, the flow is slower, more turbulent and deeper.

The spatially rapid transition from supercritical to subcritical flow is accompanied by a so-called hydraulic jump. It corresponds to the conversion of kinetic energy into potential energy and turbulence, and leads to a rise in the height of the flow.

The hydraulic analysis of the Marinada is done in Sect. 3.3.

## 3 Results

### 3.1 Spatial and temporal evolution of the Marinada at the surface

The current study uses the SMC AWS observations for this purpose. It compares the spatio-temporal evolution of the model's wind, temperature and humidity variables at the surface with the observations. According to the temporal evolution of the Marinada, the day can be divided into four phases. These four phases are shown in Fig. 4 and in the Appendices A2 and A3, and are classified as follows:

- *Sea breeze*: This phase lasts from 08:00 UTC to 16:00 UTC on 16 July 2021. During this period, a typical sea breeze develops. Along the coast and near the surface, the wind blows from west to north at night. After sunrise it turns south in the morning and southwest in the afternoon, like for the Barcelona sea breeze (Redaño et al., 1991). This breeze is associated with the arrival of a cooler (Fig. 4) and more humid air mass (Appendix A3).

  The configuration of the slope in the Alt Camp sub-region is very likely to contribute to the acceleration of the wind inland through the upslope flow mechanism, as in the Valencian region studied by Miao et al. (2003). Meanwhile, the wind in the Segre sub-basin is west to southwest, corresponding to the upslope wind direction on this side of the CPR and the westerly conditions. The temperature increases continuously in this area. The boundary between the marine and continental air masses, i.e. the sea breeze front, gradually moves inland during the morning and early afternoon until it is over the CPR at 16:00 UTC. The wind speed at 10 m a.g.l. is between 4 and 6 m s$^{-1}$.

- *Marinada onset*: Between 16:00 UTC and 20:00 UTC the marine air mass carries on its progression into the Ebro Basin as shown in Fig. 4b and Appendix A3. The marine air mass in the Ebro basin and the wind associated with it is called the Marinada. The advance of the Marinada front is associated with stronger winds than in stage 1. Temperatures decreases and humidity increases rapidly behind the Marinada front. The temperature at Els Plans have decreased by 4 K and the humidity increased by 6 g kg$^{-1}$ between 18:30 and 19:30 UTC, i.e. before and after the arrival of the Marinada (Fig. 12). The model reproduces the observed patterns, but with a slightly warmer and drier marine air mass. At this stage it can be seen that the orography plays a major role as an obstacle to the progression of the marine air. The marine air





mass enters the Ebro Basin preferentially over areas of low elevation. This study focuses on the entrance of the Conca de Barberà, where the Marinada front is the more easily distinguishable. A transect from Torredembarra to La Cendrosa is used later in this article to analyze the progression and behavior of the marine air mass. This transect also passes by Serra de Miramar, Serra del Tallat and Els Plans.

– *Mature Marinada*: From 20:00 UTC to 22:00 UTC, the front is no longer distinguishable at the surface as seen in Fig. 4c. The fresh and moist marine air mass has reached the whole of the low Segre sub-basin. The temperature decrease during this phase corresponds to the normal radiative cooling of an evening transition, and the specific humidity remains constant. The wind over the Segre sub-basin is well established with an average south-southeasterly direction.

– *End of Marinada*: From 22:00 UTC, the wind over the irrigated area starts to veer east-northeast. This is the normal direction of a downslope wind coming from the northeastern part of the CPR towards the bottom of the basin. The effect of the marine air mass is less pronounced and the circulations that develop have the characteristics of katabatic winds that follow the local slopes. These local winds have already been described and studied in this region by Martínez et al. (2008) and Cuxart et al. (2012). In the meantime, a land breeze is generated on the coast.

For each of the four phases, the model is well able to reproduce the observed patterns of near-surface temperature, humidity and wind. This makes it possible to rely on the models to study in detail the processes involved in the life cycle of the Marinada.

### 3.2 Processes involved in the Marinada event under the influence of westerlies (16 July 2021)

The simulation results are used in the following to investigate the physical mechanisms involved during the Marinada in more detail. This section focuses on 16 July 2021, a day with westerly winds, i.e. the most common weather situation encountered in the region.

#### 3.2.1 The Sea breeze phase (8:00 - 16:00 UTC)

During the first stage of the day, the marine air mass is progressively brought over the CPR. Since the model fairly reproduces the behavior observed, it can be used to explore in depth the processes at play during this first phase. To this end, the momentum source terms influencing the wind components, i.e. the acceleration terms, can be retrieved from the model. These acceleration terms are shown in Fig. 5 for 16 July 2021 at 12:00 UTC. These acceleration terms are averaged spatially according to the polygons defined in fig. 2 and Table 1, and vertically between 50 and 300 m a.g.l. This height range is selected in order to capture the acceleration terms in the well-mixed ABL. It allows to evaluate the role of the different processes on the dynamics of the marine air mass. This figure shows that the dominant acceleration term is the PGF, followed by the parameterizations for mass flux and vertical turbulence. By studying the value and direction of the PGF term, it is possible to highlight the main process at work in each sub-region.

Over the sea, advection dominates and the other terms are small. The sea acts as a provider of fresh and moist air to the lower layers of the atmosphere, but does not significantly influence the wind.





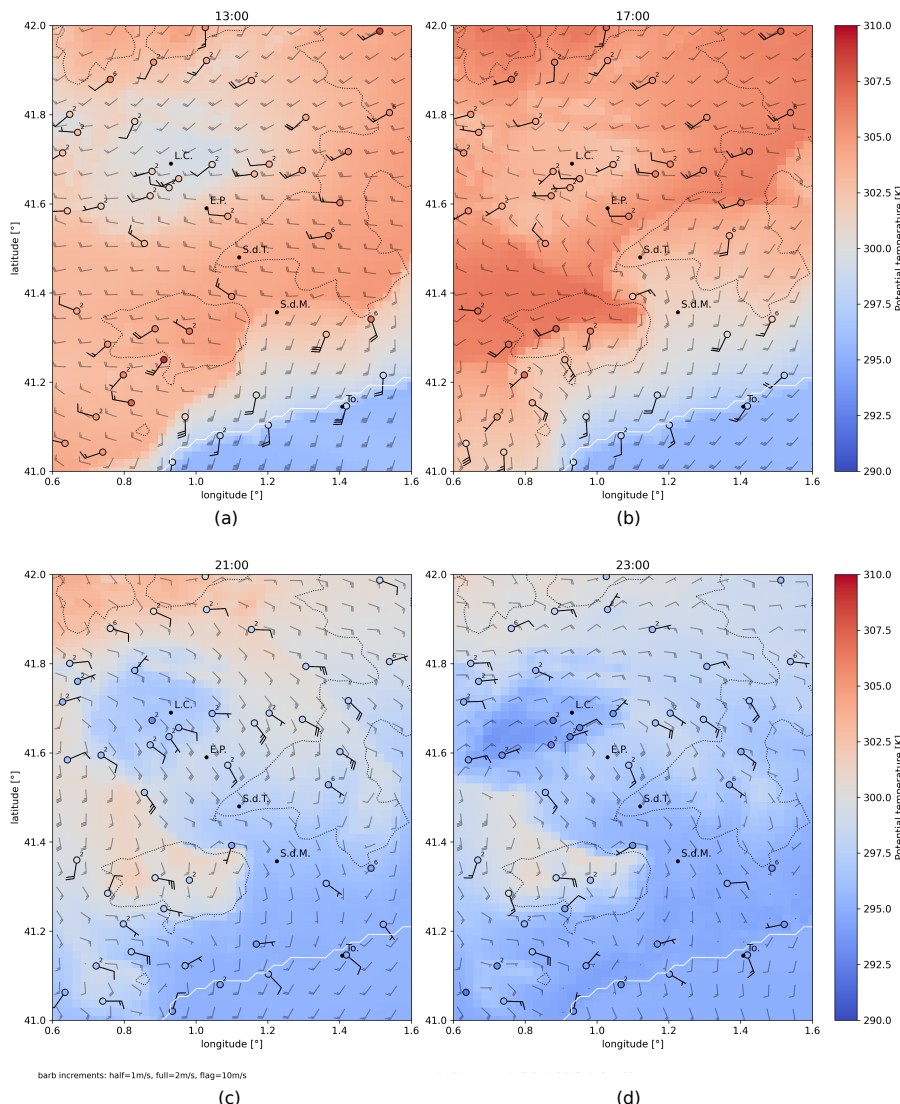

**Figure 4.** Maps of observed and modeled potential temperature and wind along the course of 16 July 2021 for the four stages of the Marinada. At 13:00 UTC is the sea breeze stage (a), at 17:00 UTC is the Marinada onset (b), at 21:00 UTC is the mature Marinada (c), and at 23:00 UTC is the Marinada decay. The color map represents the potential temperature at 2 m a.g.l., and the semi-transparent wind barbs represent the 10 m a.g.l. wind speed modeled by Meso-NH. The observations are taken from the SMC network and are shown with opaque barbs and colored points. For each station the color inside the circle represents the potential temperature measured on the same scale as for the model color map. The measured winds are shown at the available measuring height. When this measuring height is not 10 m, the actual height is indicated as a subscript. The thin dotted line represents the iso altitude line at 600 m. The acronyms L.C., E.P., S.d.T., S.d.M., and To. stand respectively for La Cendrosa, Els Plans, Serra del Tallat, Serra de Miramar, and Torredembarra.



On the coast and in Alt Camp, the PGF value is high, tends to accelerate the air inland, and is the main process at play to explain the wind tendency. In the coastal area, the direction and intensity of the PGF are typical of the sea breezes in this region (Cuxart et al., 2014). It confirms that the marine air mass is first brought to land by a sea breeze. In Alt Camp, the PGF is also the main term at play and is directed uphill, towards the north. This is typical of the PGF of an upslope wind (Serafin and Zardi, 2010). This upslope acceleration is in the same direction as the sea breeze acceleration, so the two terms merge.

In Conca de Barberà, no term dominates the others, and the terms are globally low. The orientation of this valley is mainly on a southwest-northeast axis, orthogonal to the propagation of the Marinada, so the valley slope winds are unlikely to influence the Marinada.

On the northwest slopes and in the dry area, a medium PGF is oriented towards the southwest, opposing the Marinada progression. This PGF is oriented upslope and is also orthogonal to the irrigated-dry land heterogeneity. As for the Coast and Alt Camp areas, the PGF is a combination of the surface heterogeneity and the upslope acceleration. Contribution of irrigation in this PGF is actually explored and detailed below in Sect. 3.5.

The irrigated area behaves like the sea, with low acceleration terms. It acts as a provider of fresh and humid air for low atmospheric layers but does not significantly influence the wind.

It can be seen that the PGF terms are somewhat symmetrical in relation to the mountain range. The mass flux and vertical turbulence terms are systematically in the opposite direction to the PGF term. These two terms represent the vertical mixing of scalar and vector variables between layers of the atmosphere. Since the sea-breeze has a maximum wind speed between 20 and 50 m a.g.l., which decreases with altitude in the ABL above, vertical mixing spreads the wind momentum upward, and acts as a globally decelerating term between 50 and 300 m a.g.l..

To investigate the vertical structure of the PGF and marine air mass, cross-sections can be plotted along the transect La Cendrosa - Torredembarra as shown in Fig. 6. It shows that the marine air mass entering inland has properties of a sea-breeze, with significant low level winds and a cooler potential temperature close to the shoreline (Torredembarra) than inland (Serra de Miramar). The PGF properties described above are found in Fig. 6 as well, with the symmetry in relation to the mountain range. The ABL top, characterized by the uniformity of potential temperature, is about 800 m a.g.l. in the Ebro basin, and reaches an altitude of 1500 m above Serra de Miramar. This high ABL top is facilitated by the wind convergence at the front of the marine air mass.

During the next hours of this first stage, the marine air mass carries on its progression over the CPR, helped by the sea-breeze and by the upslope wind, until the front arrives at Serra del Tallat. From this moment on, the Marinada begins.

### 3.2.2 The Marinada onset (16:00 - 20:00 UTC)

The first phase brought the cold and humid marine air mass over the CPR. Although this air mass was heated by the surface as it moved inland, it is still cooler than the surrounding air, with a virtual potential temperature between 302 and 304 K over the CPR, as can be seen in Fig. 7. In the Ebro basin, the air has been heated throughout the day, and the virtual potential temperature at 17:00 UTC is higher, around 305 K. Since the density is inversely proportional to the virtual potential temperature, the marine air mass is also denser than the Ebro basin air mass.

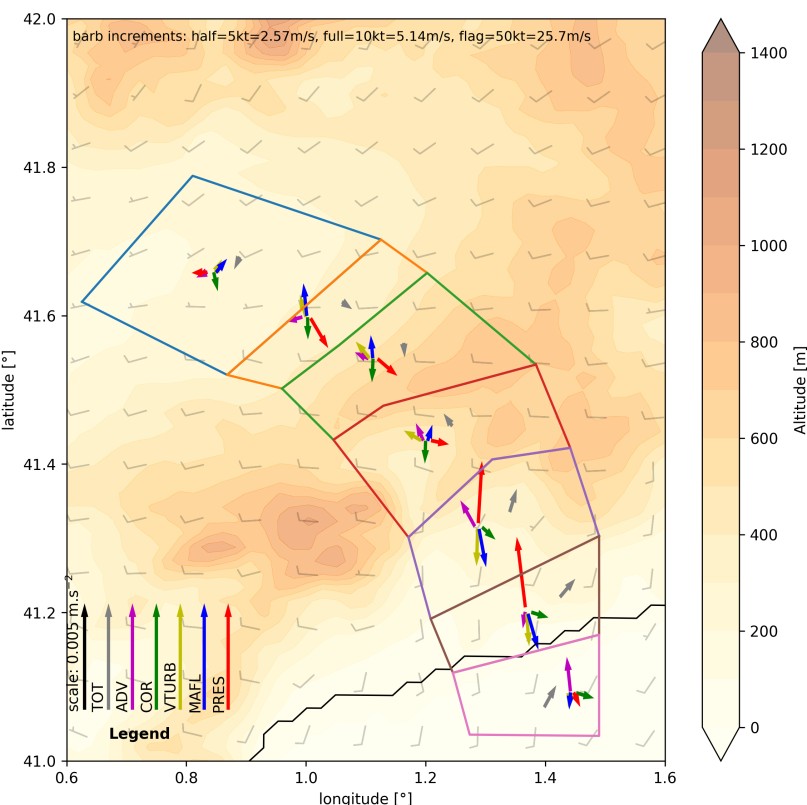

**Figure 5.** Acceleration terms of horizontal winds for 16 July 2021, at 12:00 UTC. The arrows represent the acceleration term applied to the wind averaged between 50 and 300 m a.g.l. and over the area of homogeneous features described in Table 1. The term's naming in the legend follows the naming system of Meso-NH: ADV, COR, VTURB, MAFL, and PRES correspond respectively to horizontal advection, Coriolis force, vertical turbulence, mass flux parameterization, and pressure gradient force (PGF). TOT stands for total and represents the total mean acceleration applied to the wind per area. Other terms not related to physical processes like numerical diffusion and relaxation are minor contributors and are omitted here. The semi-transparent wind barbs show the mean horizontal wind every three grid points and averaged between 50 and 300 m a.g.l. The background color map represents the relief of the region.



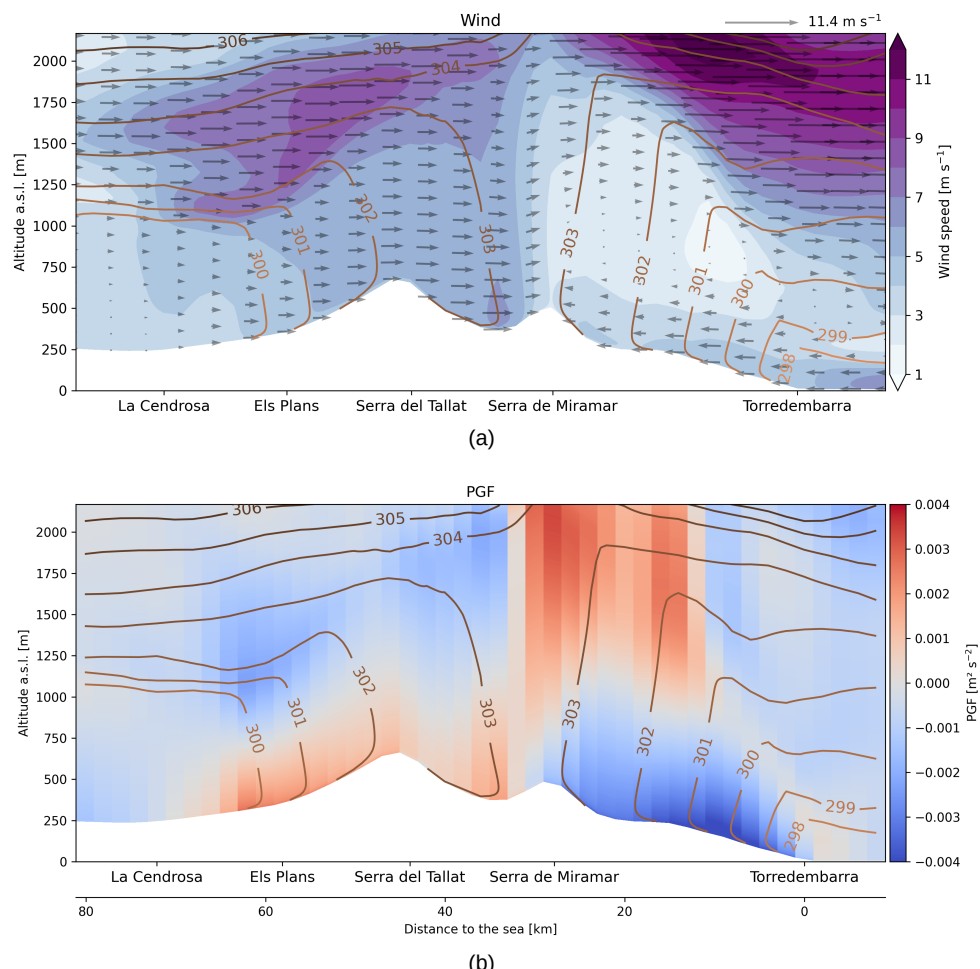

**Figure 6.** Cross-sections of horizontal pressure gradient force and wind speed on 16 July 2021, at 12:00 UTC. The colormap of (a) represents the horizontal absolute wind speed, and the arrows represent the projection of the 3D wind into the plane of the cross-section. The colormap of figure (b) is the horizontal pressure gradient force projected into the plane of the cross-section. The red color means that the acceleration is toward the right of the graph, i.e. the sea, and the blue represents an acceleration towards the left of the graph, i.e. the Ebro basin. The isolines of the two graphs represent the virtual potential temperature in kelvin.



The density gradient between the two air masses induces a pressure gradient, resulting in an acceleration towards the northwest. At 17:00 UTC, the acceleration actually takes place over Serra de Miramar and over Serra del Tallat, the boundaries of
Conca de Barberà.

The first accelerates the wind into the Conca de Barberà. The acceleration brings the wind closer to the surface and thins the ABL to a depth of 500 m. This denser layer follows the terrain up the southeastern slopes of the Serra del Tallat, generating a counter-gradient in the middle of the Conca de Barberà (Fig. 7). However, this counter-gradient is weak compared to the PGF over Serra de Miramar and only slightly slows down the wind.

Secondly, the Serra del Tallat acts like the Serra de Miramar, further accelerating the wind towards the northwest, i.e. into the Ebro basin. The result is a downslope wind on the northwest side of the CPR. The main driver of this downslope wind is the PGF, which in this case is not due to surface cooling, but to the higher density of the marine air mass. Therefore this wind belongs to the category of fall winds, and this fall wind is called Marinada.

Since the general circulation is west before the arrival of the Marinada, the boundary between the Marinada and the Ebro
basin air mass is clear. This boundary will be referred to as the Marinada front. The Marinada propagates into the Ebro basin at a speed of about 1.5 to 3 m s$^{-1}$ from 16:00 UTC to 20:00 UTC (Appendix A2). The propagation of the front is relatively slow compared to the wind speed inside the Marinada. This slow propagation is due to the counteracting westerly winds, which cause significant convergence at the Marinada front and convert some of the near-surface horizontal momentum into vertical momentum in an updraft. The vertical wind speed in the frontal updraft at 500 m a.g.l. ranges from 0.4 to 0.6 m s$^{-1}$ between
16:00 and 18:00 UTC. By 20:00 UTC the Marinada has reached the whole of the Segre sub-basin, i.e. the western part of the Ebro basin, and the Marinada enters its third phase: the mature phase.

### 3.2.3  The Mature phase of the Marinada (20:00 - 22:00 UTC)

During this phase the Marinada blows continuously throughout the whole of the Segre sub-basin. As shown in Fig. 8, the wind acceleration is still partly due to the acceleration over Serra de Miramar, but mainly to the component over Serra del Tallat
and its northwestern slope. Due to the higher density of the Marinada air mass, the cold air mass and its wind remain close to the ground. Acceleration over the northwest slope follows the same behavior and PGF values are particularly high in the lowest 200 m a.g.l. The thinning and acceleration of the marine air flow corresponds to the transition from the subcritical to the supercritical regime from a hydraulic analysis perspective. This hydraulic perspective will be developed in the next section.

The radiosonde released from Els Plans at 21:00 UTC, shown in Fig. 9, confirms the main features simulated by the Meso-
NH model. In particular, the jet of the Marinada is clearly visible, located at 70 and 100 m a.g.l. in the observations and model, and with maximum wind speeds of 9.2 and 9.3 m s$^{-1}$, respectively. This jet height and strength is similar to the Santa Ana fall winds (Jiang et al., 2022). The potential temperature profile shows a uniformly cold air in the first 100 m a.g.l., ruling out the idea that this wind is driven by a surface cooling effect as in katabatic winds. The radiosounding also shows a clear wind veering in the lower 500 m, with a south-southeast orientation and a sharp transition to the residual layer above, where the
general westerly circulation is present. The model shows a smoother transition from the Marinada layer to the residual layer between 250 and 500 m a.g.l. The wind direction changes smoothly from south to west, and the wind speed is overestimated




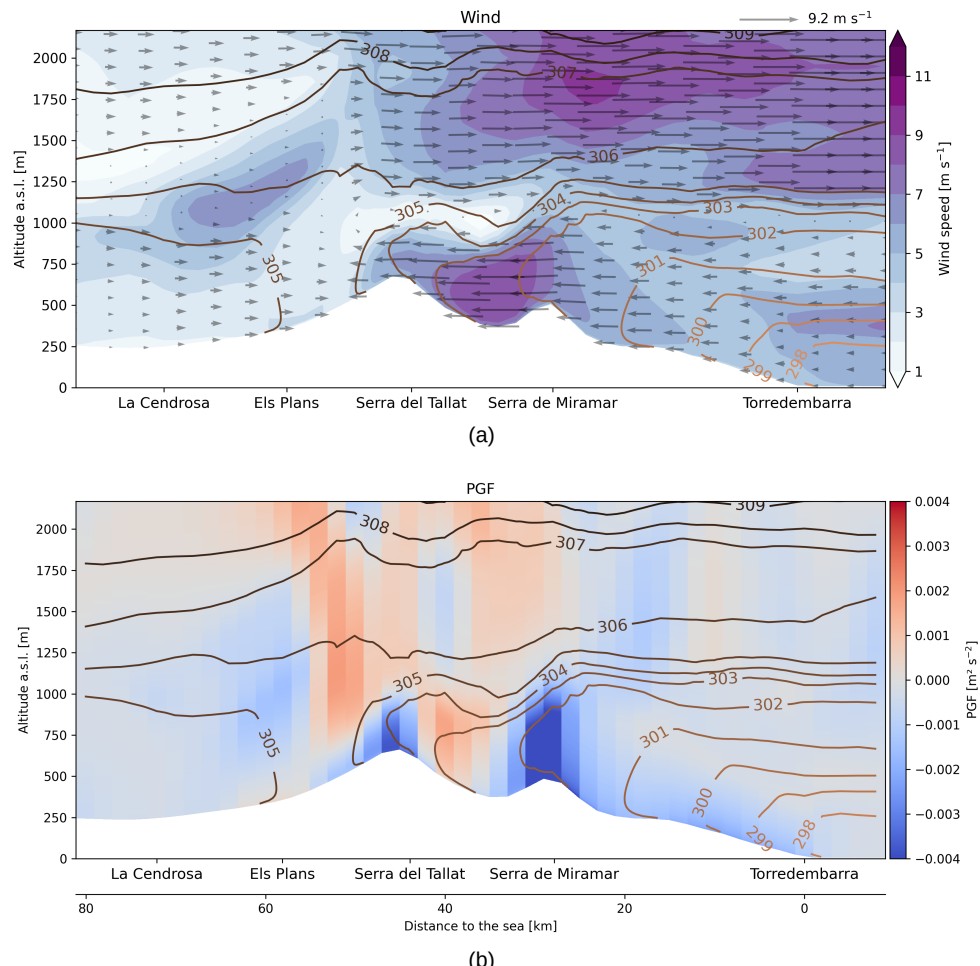

**Figure 7.** As in Fig. 6, but for 17:00 UTC

by the model above 200 m, with a modelled wind speed of about 3 m s$^{-1}$ versus an observed wind speed of about 1 m s$^{-1}$. This is probably due to an overestimated entrainment phenomenon at the entrainment zone between the Marinada air mass and the residual layer. The modelled and observed potential temperature profiles are in good agreement with respect to the depth

of the cold air mass and the stratification of the residual layer (Fig. 9a). The vertical potential temperature gradient between the Marinada and the residual layer is slightly underestimated by the model, resulting in a weaker temperature inversion cap and hence a thicker entrainment layer. The specific humidity profile is less accurately reproduced by the model, although the moisture signature of the Marinada is present in both the observations and the model. Comparing the characteristics of the Marinada at 50 m a.g.l. and the Ebro basin air mass at 250 m a.g.l. with the radiosonde observations, the Marinada is 5 K

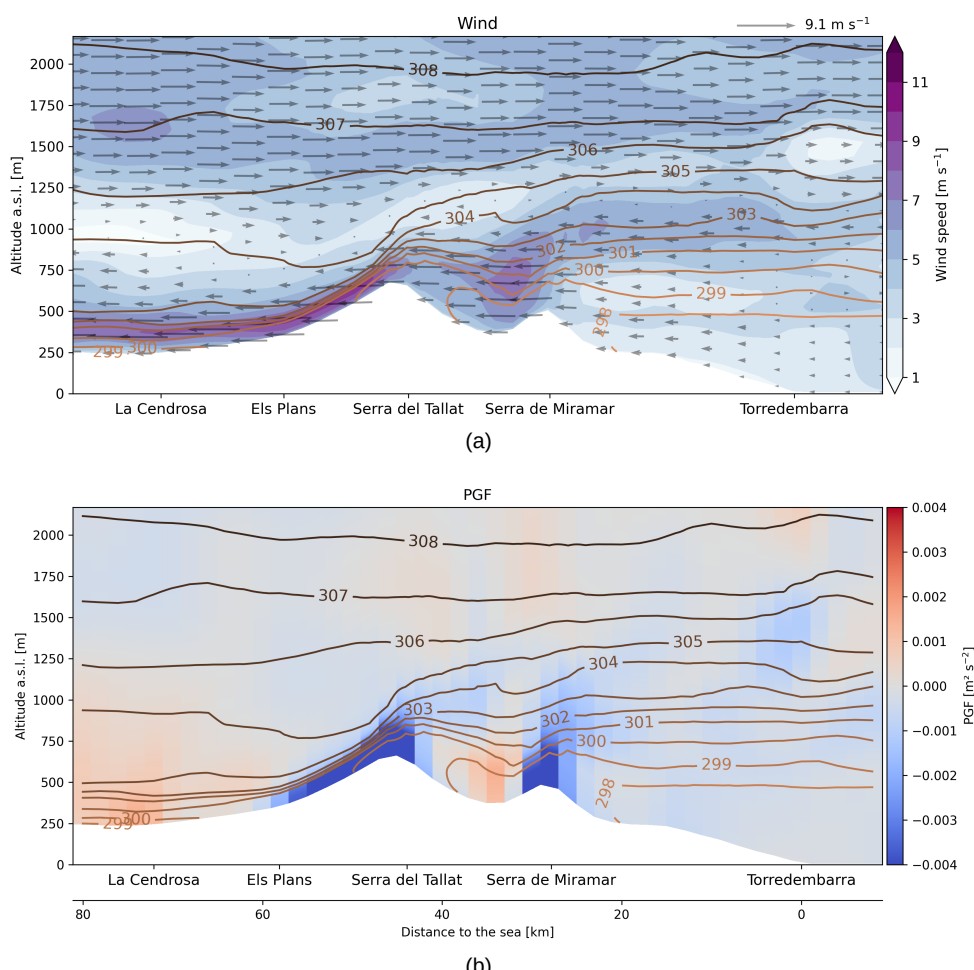

**Figure 8.** As in Fig. 6, but for 21:00 UTC

cooler and 4 g kg$^{-1}$ more humid than the Ebro basin air mass in which it flows. The model shows a temperature and humidity
difference of 4 K and 2 g kg$^{-1}$ respectively, and reproduces the wind in the lowest layer particularly well.

### 3.2.4  The Marinada decay phase (22:00 - 24:00 UTC)

During the mature phase of the Marinada, the air mass from the Segre sub-basin is gradually replaced by cooler and wetter air.
This cool advection, combined with the surface radiative cooling, decreases the air potential temperature in the Segre sub-basin.
For instance at 50 m a.g.l. above La Cendrosa, the air temperature is reduced by 4.5 K between 20:00 UTC and 22:00 UTC. At
22:00 UTC, this cooling effect begins to affect the dynamics of the Marinada. Figure 10 shows a rise in the virtual temperature
isolines over La Cendrosa, an indication of the air mass cooling. This cooling leads to a reduction in the PGF, which until then





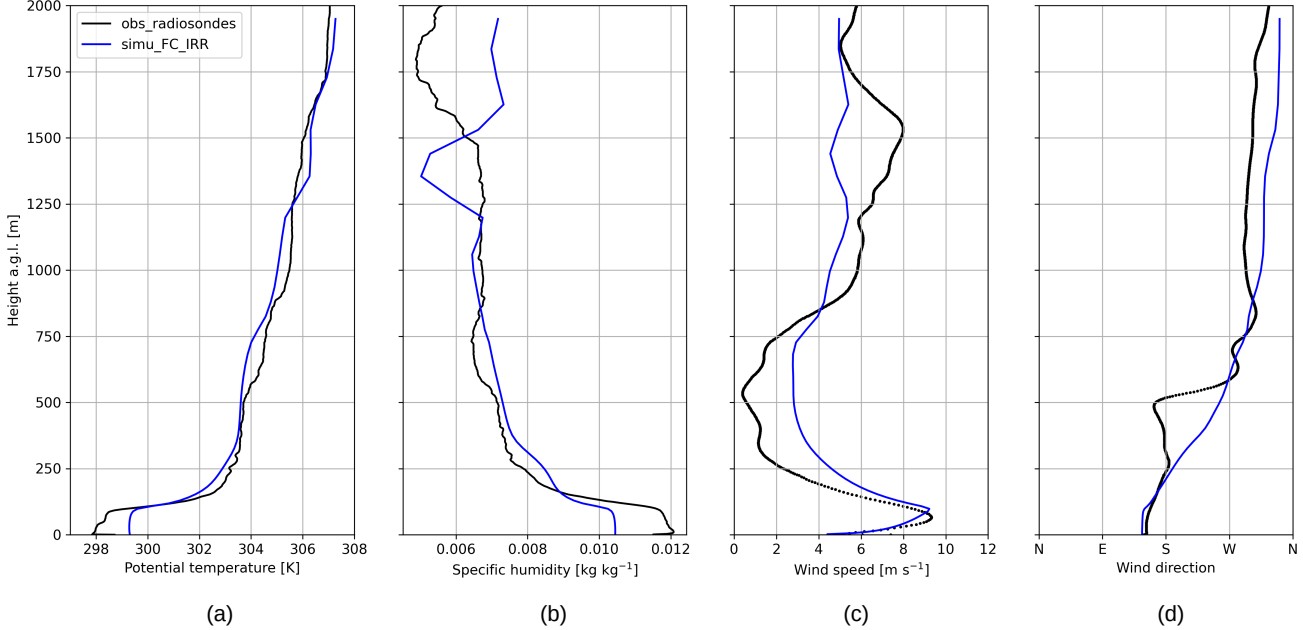

**Figure 9.** Vertical profiles of potential temperature (a), specific humidity (b), wind speed (c), and wind direction (d), above Els Plans on 16 July 2021 at 21:00 UTC. In black are the observations from the radiosonde released from the ground at 21:00 UTC. In blue are the model vertical profiles.

had accelerated the wind towards the northwest, and even induces a PGF opposite to the Marinada in the lower part of the Segre sub-basin, thus slowing down the Marinada.

At 22:00 UTC, the surface begins to be cooler than the marine air mass over Els Plans (not shown). The resulting nocturnal cooling of the lower atmosphere now contributes to the downslope PGF and helps to maintain the Marinada over the northwestern slopes of Serra del Tallat.

     Nevertheless, the opposite PGF in the low Segre sub-basin is stronger than the downslope PGF, and the Marinada is progressively confined to the northwestern slope of Serra del Tallat (Fig. 10) at 23:00 UTC, until being replaced by weaker local 390   katabatic winds around midnight (not shown).

### 3.3   Hydraulic theory perspective

A fall wind such as the Marinada has most of the characteristics of an open channel flow, and the hydraulic theory can be used to analyze it, as mentioned in the introduction. The Froude number described in Sect. 2.5 is used below for this purpose.

     The jet height associated to the Marinada and Froude number are plotted along the Torredembarra - La Cendrosa transect in 395   Fig. 11.





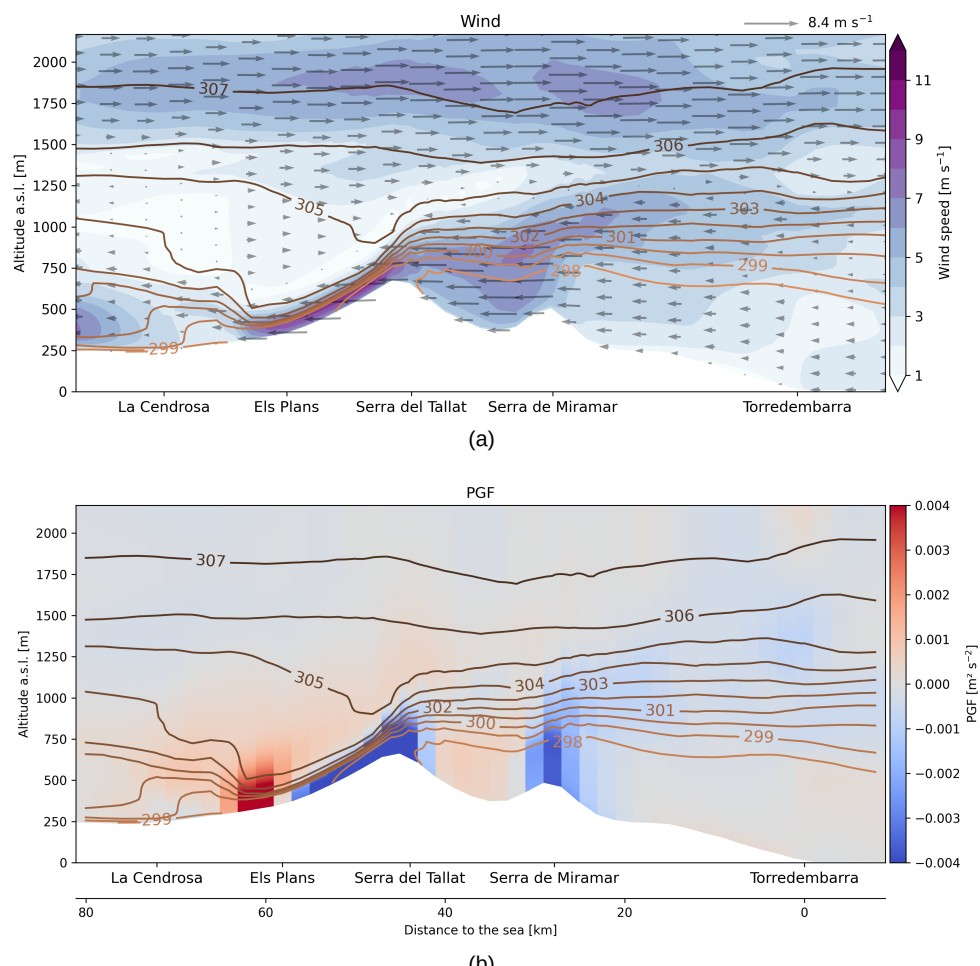

**Figure 10.** As in Fig. 6, but for 23:00 UTC

This figure shows that the jet height diagnostic $h_{top}$ described in Sect. 2.5 captures well the jet over the CPR and the Ebro basin, and does not interpret the sea-breeze east of Serra de Miramar as a jet, which was one of the purposes of the definition considered for $h_{top}$. Note that the jet height diagnostic detects two jets over Torredembarra: a residual sea breeze at 17:00 UTC, and a very weak jet on 17 July at 01:00 UTC. Neither of these jets is of interest for the Marinada study.

At all times, the jet forms to the west of Serra de Miramar. Between Serra de Miramar and Serra del Tallat, i.e. in the Conca de Barberà, the jet gets progressively closer to the surface. The jet then reaches its minimum height on the northwestern slope of the CPR, before thickening in the lower part of the Segre sub-basin.

During the Marinada onset phase, at 17:00 UTC, the Froude number ranges from 1 to 12, indicating that the jet flow is supercritical, from Serra de Miramar to the Marinada front, which located between Els Plans and Serra del Tallat at this time.



This supercritical regime confirms that Serra de Miramar plays a role in accelerating the marine air mass towards the northwest. During the mature and decaying phases, the flow over the Conca de Barberà is subcritical, which means that the flow is mainly driven by the flow behaviour downstream, i.e. the Marinada in the Segre sub-basin. The Marinada phenomenon corresponds to the supercritical flow west of Serra del Tallat. At 21:00 UTC, this flow is supercritical from Serra del Tallat to La Cendrosa. The Marinada flows freely in the Segre sub-basin. At 23:00 UTC, the Segre sub-basin has cooled down, and the cool air mass

slows down the Marinada. The height of the Marinada $h_{top}$ suddenly increases between Els Plans and La Cendrosa, and the Froude number drops below 1, meaning that the flow becomes subcritical. This corresponds to a hydraulic jump. As the Segre basin becomes cooler, the subcritical zone gradually extends to the southeast, and the Marinada is confined to the northwestern slope of Serra del Tallat. The supercritical regime associated with the Marinada at all stages confirms the gravity-driven nature of the Marinada.

The flow characteristics are also shown at 01:00 UTC on 17 July. At this time, the Froude number reaches a maximum of 3 and the jet height diagnostic does not capture the jet on all of the northwestern slopes. This circulation is a remnant of the Marinada mixed with a katabatic wind due to surface cooling. In this article it is considered that this circulation is no longer the Marinada.

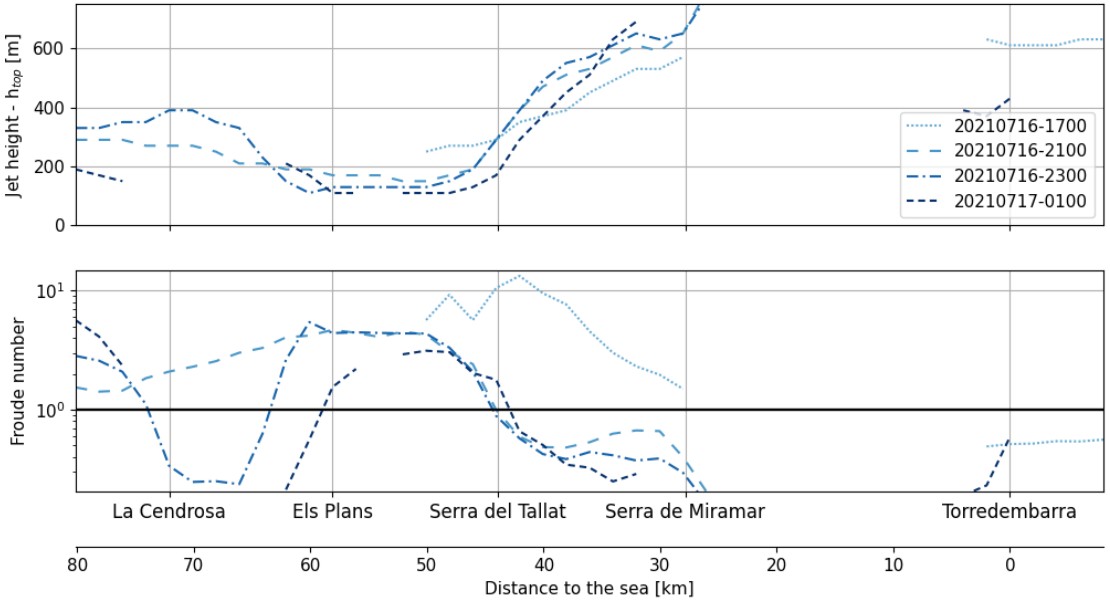

**Figure 11.** Marinada height and Froude number on 16 July 2021 at different times of the Marinada



### 3.4 Processes involved in the Marinada event under the influence of thermal low (21 July 2021)

The processes at play on 21 July are globally similar to those on 16 July. The PGF corresponding to the sea-breeze and to the upslope winds are very similar in terms of direction and magnitude. Nevertheless the general circulations over the region lead to differences in the characteristics of the Marinada. At 500 hPa the pressure gradient is weak over the Iberian peninsula, and the wind blows from the west. Low tropospheric large scale winds do not have a prevailing direction as for 16 July. Heating of the Iberian peninsula is enhanced by the absence of large-scale advection, and ascending vertical motions are responsible

for the formation of a thermal low located southwest of the Ebro basin. This thermal low generates a southeast to northwest circulation in the first lower kilometer of the troposphere in the Ebro basin. This is the direction of the Marinada and this circulation therefore helps the progression of the marine air mass into the Ebro basin. The first phase of the Marinada is therefore not mainly driven by the sea breeze as on the 16th, but also by the circulation induced by the thermal low.

This causes the marine air mass to arrive earlier over the CPR than on 16 July 2021. The Marinada starts to flow into the

Ebro basin starting around 12:00 UTC in the model, and around 13:30 UTC in the observations.

Figure 12 shows the comparison of 16 and 21 July for the evolution of wind, potential temperature, and specific humidity close to the surface over the two days. On the 21 July the model performs less well than on 16 July for the arrival time at Marinada, resulting in differences in terms of wind, potential temperature, and specific humidity throughout the day.

On 21 July, the wind direction before the arrival of the Marinada is northwest at Els Plans. This northwesterly direction,

opposite to the southeasterly general circulation, is due to a local mesoscale circulation, probably induced by the irrigated–dry heterogeneity. However, this irrigation breeze is not strong enough to counteract the arrival of the Marinada, and the veering to the southeast finally allows the arrival time of the Marinada to be estimated. As the general circulation does not counteract the Marinada, the wind speed is higher on 21 July than on 16 July. The potential temperature is also globally higher on 21 July than on 16 July. This is one of the conditions for the development of the thermal low (Jimenez et al., 2023). Also, since the

Marinada arrives in the middle of the day on the 21st, its arrival coincides with the strong daytime surface warming, and the Marinada air mass is warmed and mixed with higher layers as it descends into the Segre basin. This mixing effect influences the evolution of temperature and humidity variables throughout the day. In particular, the arrival of the Marinada on the 21 July does not lead to such a sharp decrease (increase) in the potential temperature (specific humidity) as on the 16 July. The Marinada onset phase lasts between 12:00 UTC and 16:00 UTC on 21 July.

Differences in the Marinada characteristics between 16 and 21 are also found in the vertical structure during the mature phase of the Marinada, as shown in Fig. 9 and Fig. 13. The mature phase is between 16:00 UTC and 20:00 UTC on 21 July. The wind direction profile shows that there is no westerly circulation in the lower troposphere. Also, according to the potential temperature profile, the troposphere is neutral over the Marinada and up to 1400 m a.g.l. on 21 July, whereas the troposphere is more stratified on 16 July. This neutral stratification is probably a residual of the ABL present prior to the arrival at Marinada.

The potential temperature gradient between the Marinada and the overlying direct layer is about 3 K on the 21 July, compared to 5 K on the 16 July. This weaker potential temperature cap, combined with the still active thermal convection, results in a deeper Marinada on 21 July. The height of the Marinada, diagnosed by the jet height $h_{top}$ described in Sect. 3.2.4, is 365 m



**Figure 12.** Time series of wind speed (a), wind direction (b), specific humidity (c), and potential temperature (d) for the 16 and 21 July 2021 at Els Plans. In black are the observations and in blue are the simulation outputs. All data are considered at 10 m a.g.l.



and 450 m, respectively, according to the observation and the model. Despite the higher jet height $h_{top}$, the Froude number diagnostics show very similar behaviours between 16 and 21 July for the onset and mature phase of the Marinada, i.e. the flow

changes from subcritical to supercritical as it passes Serra del Tallat (not shown). There is also a hydraulic jump at the end of the mature phase at 20:00 UTC between La Cendrosa and Els Plans (not shown). This allows to confirm that the Marinada can also be classified as a fall wind on the 21 July 2021.

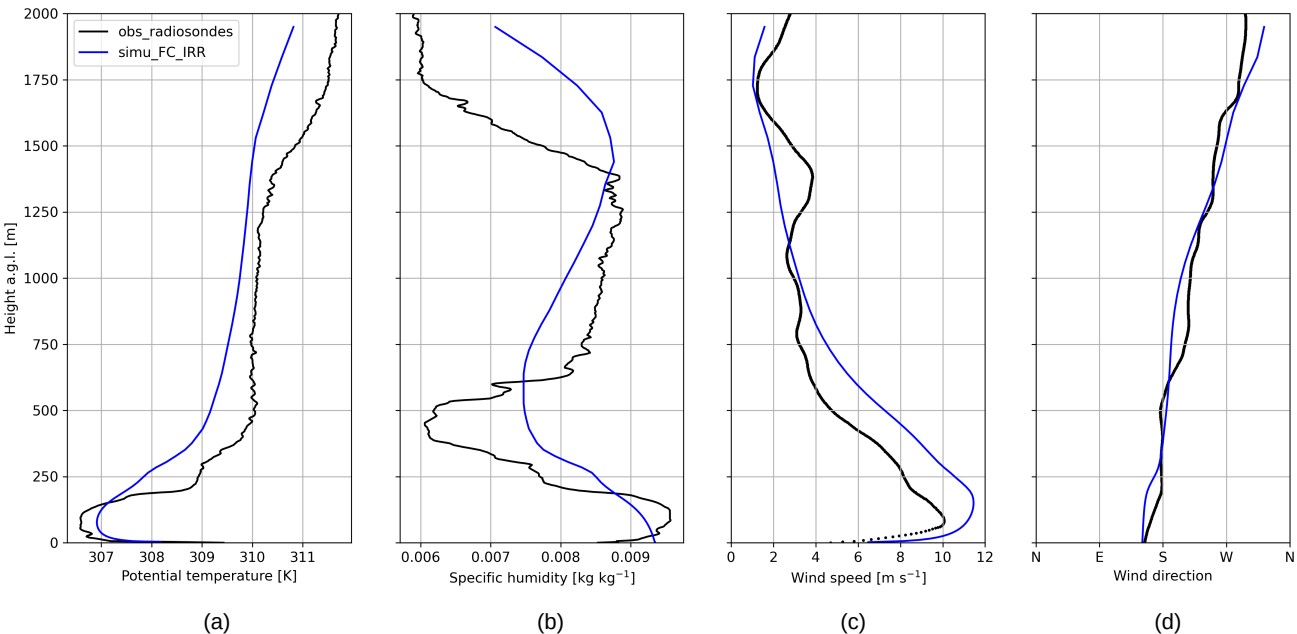

**Figure 13.** As in Fig. 9, but for 21 July 2021, at 17:00 UTC

## 3.5    Influence of irrigation

The previous sections have shown that the Marinada is a mesoscale circulation driven by the topography of the region. First,

the sea-land heterogeneity and the southeastern slopes of the CPR lead to sea breeze and upslope wind circulations that bring a cool and moist air mass over the CPR. The north-western slope of the CPR then accelerates the dense marine air mass into the Segre sub-basin. Secondly, it has been shown that the characteristics of the Marinada also depend on the strength of shallow convection due to surface heating, which in turn depends on the timing of the Marinada onset and the presence or absence of westerlies. These influencing surface features are natural. Another topographic feature is important to consider and should be

studied further as it is anthropogenic: irrigation.





### 3.5.1 Influence on PGF and air density

To investigate the influence of irrigation, the three different model configurations described in Sect. 2.4.2 are run for 16 July. Figure 14 shows a comparison of the results for PGF and potential temperature in the well-mixed boundary layer, i.e. between 50 and 300 m a.g.l. As irrigation is mainly concentrated in the Ebro basin, its effect is mainly found to the west of the CPR.

During the first phase, at 12:00 UTC, the PGF due to irrigation heterogeneity and the slope acceleration can be dissociated thanks to the non-irrigated run NOIRR. It is found that most of the positive PGF, i.e. the acceleration towards the southeast, on the Ebro basin side is due to the irrigated–rainfed heterogeneity. The boundary between irrigated and rainfed is between La Cendrosa and Els Plans, where the PGF is maximum at 12:00 UTC. This PGF is about 1.5 mm s$^{-2}$ and decreases progressively towards the east until it reaches 1.0 mm s$^{-2}$ at Serra del Tallat. On the southeast side of the CPR, the PGF is symmetrically

opposite, the absolute PGF has a maximum of 3.5 mm s$^{-2}$ and decreases to 2 mm s$^{-2}$ at Serra del Miramar. The negative sign means that the acceleration is towards the northwest. Since the CPR slopes on both sides are about the same steepness, the upslope PGF must be of the same magnitude. Thus, similar to the Ebro basin side, the inland acceleration between Torredembarra and Serra de Miramar is mainly due to the coastal sea breeze.

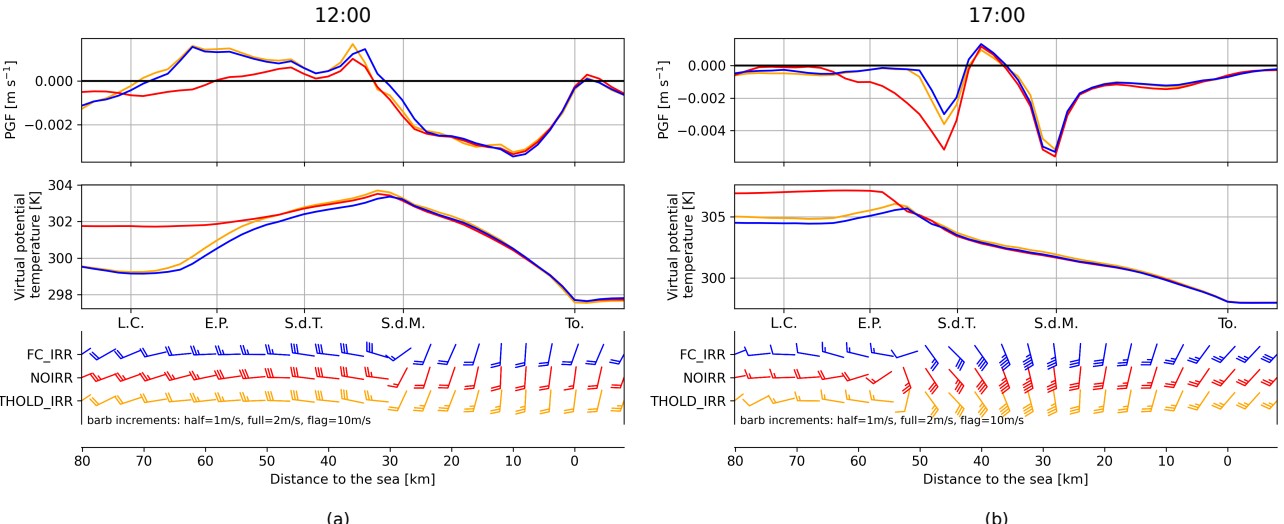

**Figure 14.** Pressure Gradient Force (PGF), virtual potential temperature and wind along the La Cendrosa – Torredembarra transect for 16 July 2021 at 12:00 UTC (a) and 17:00 UTC (b). The values are averaged vertically between 50 and 300 m a.g.l.

The potential temperature is strongly influenced by irrigation in the Ebro basin. Above La Cendrosa, irrigation reduces

the potential temperature between 50 and 300 m a.g.l. by 2.5 K at 12:00 UTC and 2 K at 17:00 UTC. The decrease in virtual potential temperature above Els Plans due to irrigation is also shown for 17:00 UTC along the vertical profile in Fig. 15a. At 17:00 UTC, the effect of the irrigation parameterization can be seen. Since THOLD_IRR represents a soil that dries





|  | Observation | NOIRR | THOLD_IRR | FC_IRR |
|---|---|---|---|---|
| 16 July | 19:00 | 18:00 | 18:30 | 18:40 |
| 21 July | 13:40 | 11:30 | 11:50 | 11:50 |

**Table 3.** Arrival time of the Marinada front at Els Plans, observed and modelled with the three different irrigation parameterizations. The determination of the arrival time is done with the wind veering from west (180° to 360°) to southeast (160° to 140°) for 16 July, on 10 minutes data.

periodically, the mean surface latent heat flux over the irrigated field is lower for THOLD_IRR, the sensible heat flux is higher, and the potential temperature is finally higher than for FC_IRR. This higher potential temperature is equivalent to a lower density of the air in the ABL, and it affects the PGF at the boundary between the Ebro basin and the marine air masses. For this reason, over Serra del Tallat, the intensity of the PGF at 17:00 UTC depends on the irrigation representation; no irrigation (NOIRR) or weaker irrigation (THOLD_IRR) leads to higher absolute PGF (Fig. 14).

### 3.5.2 Impact on arrival time and wind speed

The differences in PGF during the first and second phases of the Marinada then induce effects on its characteristics. The irrigation PGF during the first phase induces a higher wind speed for the westerly winds. These winds slow down the arrival of the marine air mass over the CPR and thus delay the onset of the Marinada. Table 3 shows the arrival of the Marinada at Els Plans for the three models and for 16 and 21 July. As already discussed in Sect. 3.4, the model as a whole represents a Marinada that arrives too early in the Segre sub-basin. However, without the irrigation representation in the model, the Marinada arrives even earlier. The difference between THOLD_IRR and FC_IRR is less than the uncertainty in the arrival time determination.

To better understand the reasons why the model simulates too early Marinadas, more observations would be needed over the regions of Conca de Barberà and Alt Camp. This would allow a better characterization of the biases between model and reality in the ABL during the first phase, i.e. the sea-breeze phase (Sect. 3.2.1). Without these observations at the first phase, it is difficult to draw conclusions regarding the reasons for the difference in arrival time at the second stage.

During the Marinada onset, the PGF induced by irrigation at La Cendrosa still acts as a decelerating force for the Marinada, but the PGF due to density differences between the two air masses also plays a role. The more irrigated the Segre sub-basin is, the weaker the density PGF is, and the slower the Marinada wind speed should be. Figure 15 shows the vertical profiles at Els Plans of potential temperature before the Marinada, and of wind speed just after the arrival of the Marinada and 2h00 later, during the mature Marinada phase. It systematically overestimates the wind speed in the lower layers of the troposphere (up to 750 – 1000 m a.g.l.). The irrigated runs perform better in this sense. In the mature phase, FC_IRR have a lower wind speed than THOLD_IRR, as expected from the density PGF value of each run. In comparison with the radiosoundings, the 2 models perform similarly well, with a slight negative bias for FC_IRR and a slight positive bias for THOLD_IRR.

These results show that adding irrigation to the simulation improves the virtual potential temperature profile, timing and wind speed of the Marinada wind. It can also be assessed that the irrigation parameterization that gives the best result for the Marinada modeling is FC_IRR, i.e. a simple parameterization where the soil is always very wet. Increasing the complexity of





the irrigation parameterization by allowing the soil to dry out does not necessarily lead to a better modeling of the mesoscale meteorological circulations and scalar variables. The possible reasons for this are discussed in Sect. 4.

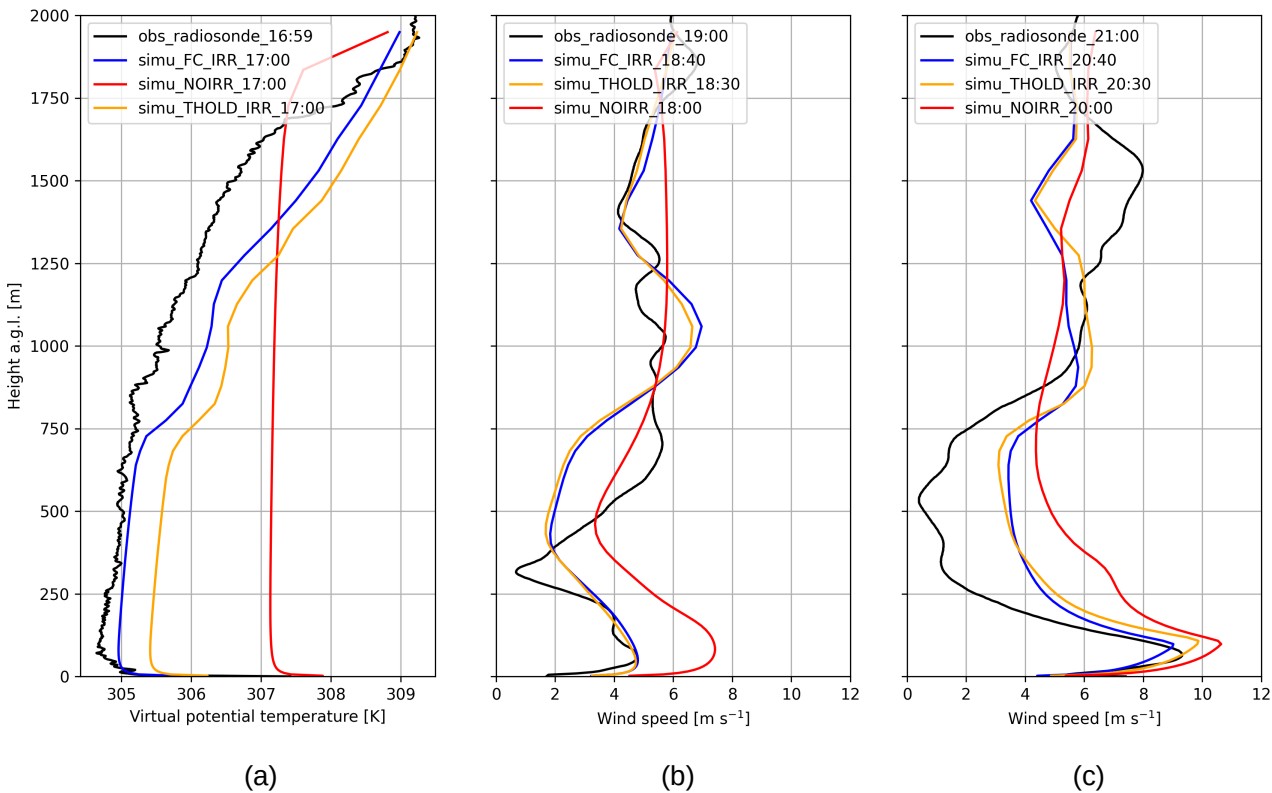

(a)       (b)       (c)

**Figure 15.** Vertical profile of the virtual potential temperature at Els Plans on 16 July 2021, before the Marinada arrival at 17:00 UTC (a), and of the wind speed right after the Marinada arrival (b) and 2h00 later (c). As the Marinada arrival is different in the model runs, the time considered is different for each model in (b) and (c), and the corresponding time is given in the legend.

## 4 Discussion

The current study focuses on understanding the Marinada, a mesoscale circulation that is common in summer in Catalonia. It uses a combination of observational and model data to elucidate the physical mechanisms involved in the formation of the

Marinada. The investigation is carried out through case studies on two days in July 2021, representing two different summer weather situations for the region. These two situations correspond to the two main weather situations in the eastern Ebro basin according to the statistical analysis of Jimenez et al. (2023) (westerlies and thermal low). The two case studies take place during the special observation period of the LIAISE campaign, which allows to validate the model outputs with a larger number of





observations. Observational data from the LIAISE campaign and from the SMC network allow to confirm the overall good
behavior of the model, providing confidence in the process analysis derived from the model diagnostics.

From the atmospheric characteristics and the wind momentum source terms obtained by the model, the main driving forces leading to the formation of the Marinada are revealed. Among the different momentum source terms, the pressure gradient force (PGF) occupies a prominent position. It is shown to be the most influential term among the others and can be related to surface-atmosphere interactions. The study divides the phenomenon into four distinct stages. In the first stage, the coastal breeze acts
together with the upslope wind to bring a cool and humid marine air mass over the Catalan Pre-coastal Range (CPR). During this time, the air mass in the Segre sub-basin is warmed. The difference in potential temperature and density between the two neighbouring air masses, assisted by the slope that descends towards the Segre sub-basin, leads to a strong acceleration in the first few hundred metres above the ground, creating the Marinada. It is also shown that, at this stage, irrigation delays the arrival time of the Marinada in the Segre sub-basin. After its arrival, the Marinada blows continuously for about 2 hours in the Segre
sub-basin. Characteristics such as height of the Marinada, wind speed, potential temperature and humidity profile are shown to be influenced by the convective activity and by the presence of irrigation. After some time, the Marinada has cooled the air of the Segre sub-basin by advection and another pressure gradient force appears, associated with a hydraulic jump and in the opposite direction to the Marinada. This PGF eventually leads to the Marinada decay. The characteristics and drivers described at each stage clearly show that the Marinada can be classified as a fall wind.
Fall winds are well known, but they are generally driven by synoptic-scale circulations and air masses, such as the Mistral, the Bora or the Santa Ana. The Marinada fall wind is specific in this sense, as it is driven by the local circulations that lead to the advection of a dense air mass over a mountain range.

Nevertheless, it shares some characteristics with fall winds. In particular, the height of the jet corresponds well with the Santa Ana wind (Jiang et al., 2022). The Marinada is influenced by the air temperature of the Ebro basin, just as the Bora is
influenced by the sea surface temperature. Specifically, the warmer the air on the lee side, the stronger the wind (Enger and Grisogono, 1998).

The influence of irrigation on the air temperature of the Ebro basin is a well-known feature of irrigation (Lobell et al., 2008; Sorooshian et al., 2011; Sridhar, 2013; Lawston et al., 2020). Interestingly, between the two irrigation parameterizations available, the simplest one gives better results compared to the radiosoundings. The fact that the Marinada is better modelled
by a simplistic irrigation parameterization such as FC_IRR than by a more realistic one such as THOLD_IRR is also not entirely new. Shrestha et al. (2018) found that this might be due to the root distribution profile in the soil in the surface model. Since SURFEX uses exponential root distributions, which are not representative of semi-arid areas such as the Ebro Basin, the plants are too sensitive to the soil moisture of the top soil layers and the surface sensible heat flux is rapidly overestimated once the soil starts to dry. Similarly, root water uptake is oversimplified in the model, for example by ignoring root hydraulic
resistance, which affects the modelled plant transpiration. The simplicity of the model and its high sensitivity to top layer desiccation means that more water has to be added in the model than is needed in reality for the modelled plants to transpire at full potential. This can explain why the rather realistic amount of irrigation water of the THOLD_IRR parameterization is not sufficient to cool the temperature adequately compared to the observation.



The framework developed here makes it possible to hypothesize a number of trends in the evolution of the Marinada over the next few decades. With climate change, land temperatures are expected to increase faster than sea temperatures in the coming decades (Sutton et al., 2007), so the temperature difference between the inland Ebro basin and the marine air mass may also increase, leading to more frequent and stronger Marinadas. This assumption is in line with the increase in the frequency of Marinada events in the last 20 years observed by Jimenez et al. (2023). Also, the heavy irrigation currently applied to the Segre sub-basin could be reduced due to the limited water resources in the region, as seen in the summer of 2023 for the first time since the construction of the Canal d'Urgell (Confederación Hidrográfica del Ebro, 2023). The reduction in irrigation, both in terms of water volume and total irrigated area, will most likely lead to stronger winds.

In summary, this study sheds new light on a local summer circulation of Catalonia, the Marinada. Until now, the physical mechanisms leading to its formation were unknown. This study shows that the Marinada can be understood as a fall wind whose cold air mass is carried over the mountain range by local circulations, i.e. sea breeze and upslope wind.

## 5 Conclusions

The current study investigated the Marinada, a local circulation of the northeastern Ebro basin in Catalonia, in northeastern Spain. This wind is important for the inhabitants of the Segre sub-basin, as it brings more windy and cooler conditions in the late afternoon in summer. The study focused on unravelling the dynamics governing its formation and spatio-temporal characteristics, thanks to the combined use of observations and simulations, in order to improve the modeling and forecasting capabilities associated with this meteorological phenomenon.

The Marinada is identified as a fall wind and distinguishes itself by the fact that the cold air mass is brought over the mountain range by local circulations. In a first stage, the sea breeze and the upslope wind act together to suck marine air onto the Catalan Pre-coastal Range (CPR). The difference in air mass density between the marine air and the warm air from the Ebro basin causes the wind to accelerate into the Ebro basin, creating what is known as the Marinada. The top of the Marinada is between 100 and 400 metres above ground level, depending on the weather and convective activity. Inside the Marinada the air is between 2 and 5 K colder than the surrounding air mass. The Marinada undergoes decay once a sufficient amount of cold air is introduced into the Ebro basin, generating a counter Pressure Gradient Force (PGF) that decelerates the Marinada. This decay process can be studied by hydraulic analysis, conceptualizing the Marinada as a supercritical flow flowing over the northeastern slopes of the CPR. The Marinada decay then corresponds to an hydraulic jump in the lower part of the Segre sub-basin. Irrigation was found to be a significant factor in the behavior of the Marinada. It affects its behavior by delaying its arrival and slowing down the internal wind speed.

This study builds a consistent framework to understand the Marinada. By doing so, this work lays a foundation for a deeper understanding of the Marinada, providing insights into its intricacies under varying weather scenarios and irrigation conditions. The knowledge gained paves the way for more accurate modeling and prediction of this phenomenon, offering valuable implications for both short-term weather forecasts and long-term climate projections.



*Code and data availability.* The observational data sets analyzed in this study are available in the LIAISE database, accessible at https://liaise.aeris-data.fr/page-catalogue/. The surface–atmosphere coupled model Meso-NH is open-source and available at http://mesonh.aero.obs-mip.fr/. The generated model output data supporting the results of this study are available from the corresponding author upon reasonable request.

*Author contributions.* T. Lunel performed the simulation, processed the experimental and model data, performed the analysis and wrote the
manuscript. M. A. Jimenez, J. Cuxart and D. Martinez-Villagrasa supervised the research and helped in interpreting the results. All authors discussed the results and commented on the manuscript.

*Competing interests.* All authors report no competing interests.

*Acknowledgements.* The authors would like to thank Josep Ramon Miró Cubells for providing the AWS data from the Servei Meteorològic de Catalunya. The authors would also like to thank the UK Met Office, and in particular Jeremy Price, Martin Best and Jennifer Brooke for
setting up the Els Plans site and providing the corresponding in-situ and radiosounding data.

This work was partially funded by the international mobility grant from the Toulouse graduate school of Earth and Space Science (TESS), and by the French component of the LIAISE project, HILIAISE, Agence National de Recherche (ANR) grant number ANR-19-CE01-0017. This work is also part of the research projects RTI2018-098693-B-C31 and PID2021-124006OB-I00 funded by MCIN/AEI/10.13039/501100011033 and by the European Regional Development Fund (ERDF A way of making Europe).



**Figure A1.** Stations location. The black dots are the SMC automatic weather stations considered in the study and gathered in Table A1. The red dots are the two LIAISE observation sites of La Cendrosa (L.C.) and Els Plans (E.P.). The green dots are reference points for locating topographical features on cross-sections. They are mountains, like Serra del Tallat (S.d.T.) and Serra de Miramar (S.d.M.), and the coastal town of Torredembarra (To.). The thin dotted line represents the iso altitude line at 600 m.



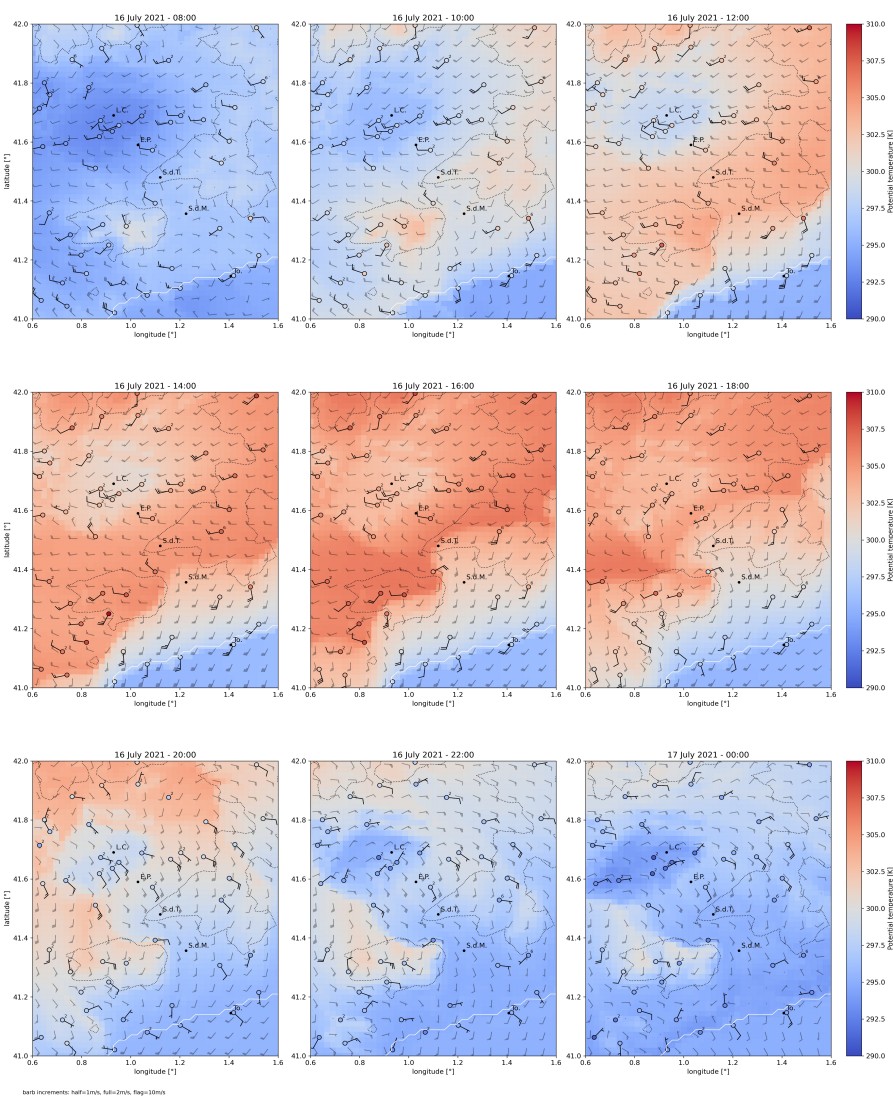

**Figure A2.** Maps of observed and modeled potential temperature and wind along the course from 08:00 UTC on 16 July 2021 to 00:00 UTC on 17 July 2021. The color map represents the potential temperature at 2 m a.g.l., and the semi-transparent wind barbs represent the 10 m a.g.l. wind speed modeled by Meso-NH. The observations are taken from the SMC network and are shown with opaque barbs and colored points. For each station the color inside the circle represents the potential temperature measured on the same scale as for the model color map. The measured winds are shown at the available measuring height. When this measuring height is not 10 m, the actual height is indicated as a subscript. The thin dotted line represents the iso altitude line at 600 m. The acronyms L.C., E.P., S.d.T., S.d.M., and To. stand respectively for La Cendrosa, Els Plans, Serra del Tallat, Serra de Miramar, and Torredembarra.



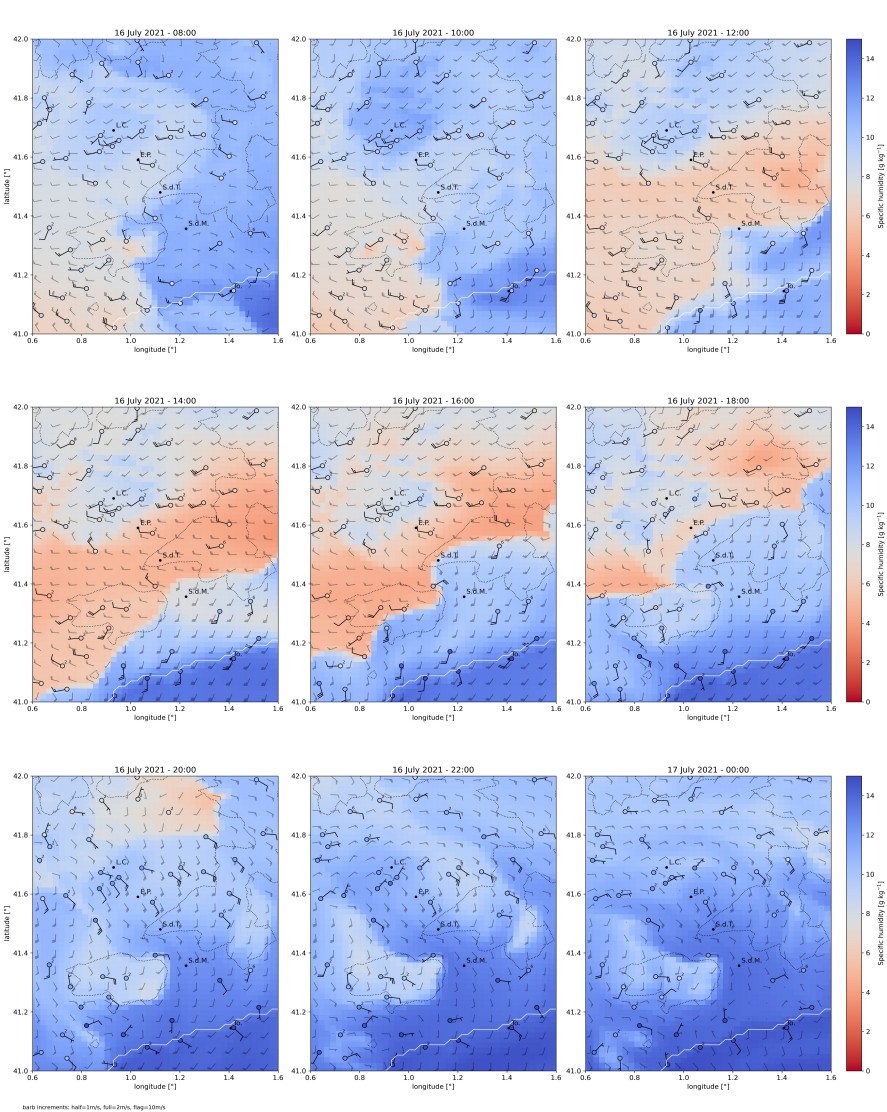

**Figure A3.** As in Fig. A2 but with specific humidity instead of potential temperature.



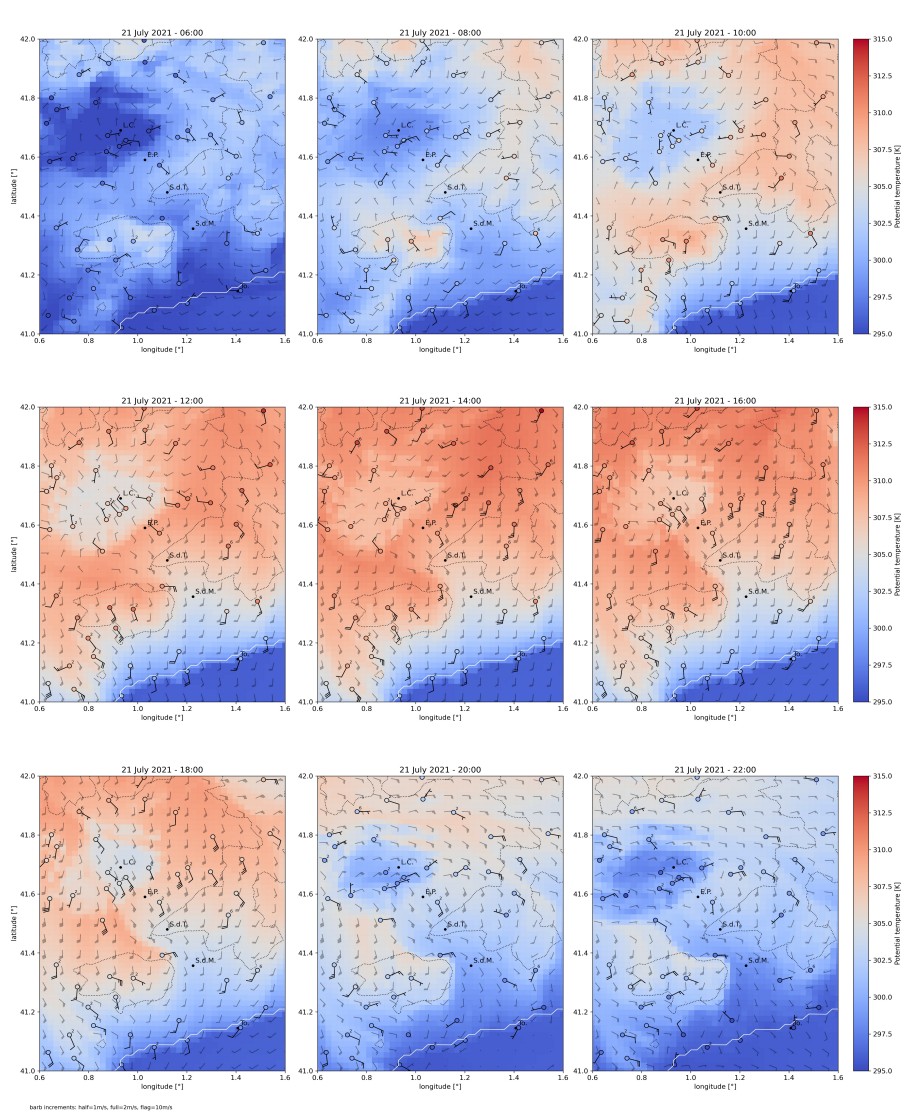

**Figure A4.** As in Fig. A2 but for 21 July 2021.





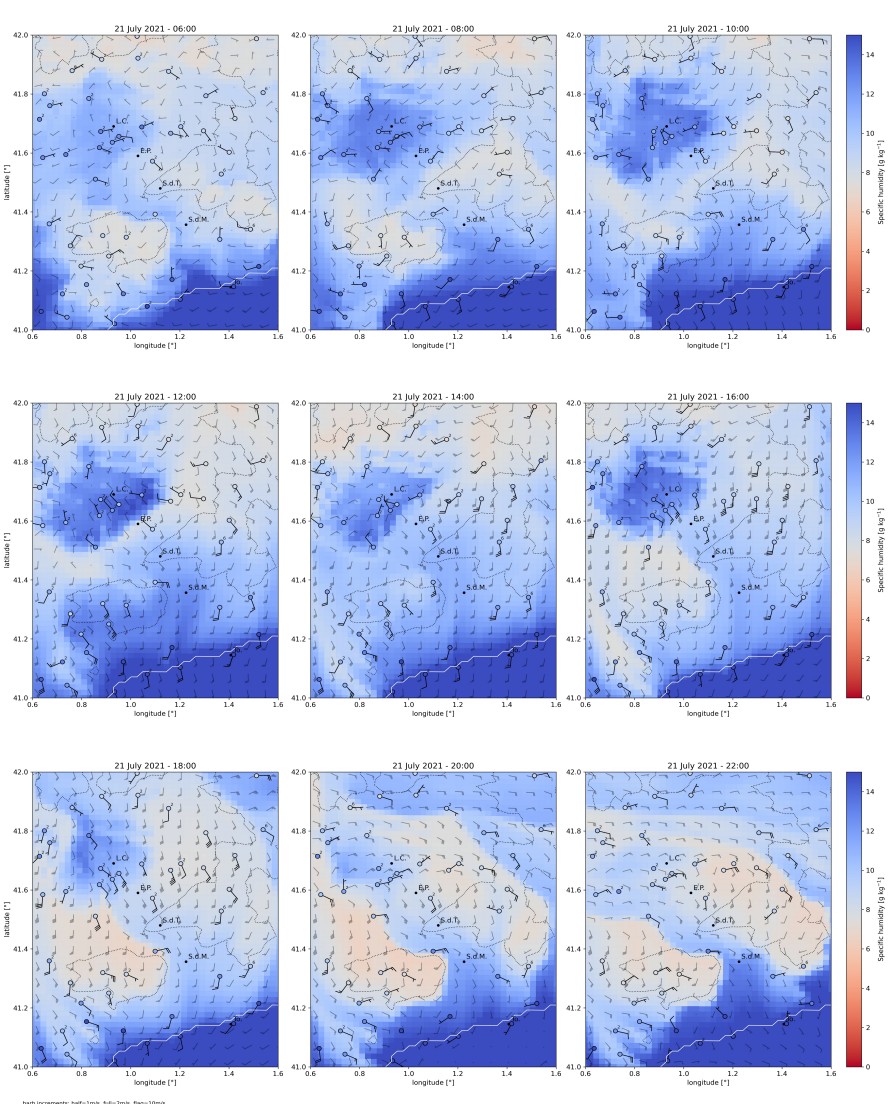

**Figure A5.** As in Fig. A3 but for 21 July 2021.




| Code | Name | Longitude [°] | Latitude [°] | Altitude [m] | Code | Name | Longitude [°] | Latitude [°] | Altitude [m] |
|---|---|---|---|---|---|---|---|---|---|
| DQ | Vila-rodona | 41.30728 | 1.36259 | 287 | C6 | Castellnou de Seana | 41.65660 | 0.95172 | 264 |
| VY | Nulles | 41.25095 | 1.29863 | 240 | V8 | El Poal | 41.67279 | 0.87741 | 223 |
| XA | Montmaneu | 41.60257 | 1.40070 | 785 | WC | Golmés | 41.63642 | 0.92446 | 261 |
| XB | La Llacuna | 41.47790 | 1.53529 | 584 | XI | Mollerussa | 41.61817 | 0.87182 | 247 |
| YH | Pujalt | 41.71734 | 1.42403 | 747 | D1 | Margalef | 41.28521 | 0.75383 | 404 |
| U6 | Vinyols i els Arcs | 41.08017 | 1.06661 | 29 | MR | Cornudella de Montsant | 41.25079 | 0.91060 | 500 |
| XR | Prades | 41.31481 | 0.98161 | 926 | WJ | El Masroig | 41.12230 | 0.72182 | 141 |
| YF | Mont-roig del Camp | 41.02066 | 0.93450 | 43 | WR | Torroja del Priorat | 41.21630 | 0.79748 | 300 |
| YL | Riudecanyes | 41.12270 | 0.96951 | 170 | XD | Ulldemolins | 41.32000 | 0.88570 | 687 |
| D9 | El Vendrell | 41.21553 | 1.52121 | 59 | X1 | Falset | 41.15374 | 0.81953 | 359 |
| UH | El Montmell | 41.34171 | 1.48769 | 545 | VB | Benissanet | 41.06289 | 0.63517 | 32 |
| WO | La Bisbal del Penedès | 41.27151 | 1.46717 | 185 | Y6 | Tivissa | 41.04343 | 0.74032 | 317 |
| CW | L'Espluga de Francolí | 41.39241 | 1.09894 | 446 | C8 | Cervera | 41.67555 | 1.29609 | 554 |
| UJ | Santa Coloma de Queralt | 41.52879 | 1.36830 | 709 | VD | Els Plans de Sió | 41.68939 | 1.20381 | 429 |
| X8 | Blancafort | 41.44237 | 1.15998 | 438 | YE | Massoteres | 41.79409 | 1.30576 | 513 |
| UM | La Granadella | 41.35991 | 0.66789 | 505 | VM | Vilanova de Segrià | 41.71450 | 0.62839 | 222 |
| YD | Les Borges Blanques | 41.51135 | 0.85617 | 283 | XM | Els Alamús | 41.59522 | 0.73507 | 235 |
| CQ | Vilanova de Meià | 41.99546 | 1.02569 | 594 | YJ | Lleida | 41.58462 | 0.64172 | 170 |
| UY | Os de Balaguer | 41.87912 | 0.76103 | 576 | VP | Pinós | 41.80483 | 1.53853 | 659 |
| V1 | Vallfogona de Balaguer | 41.78487 | 0.82939 | 238 | XT | Solsona | 41.98766 | 1.51165 | 691 |
| WA | Oliola | 41.87694 | 1.15410 | 443 | DK | Torredembarra | 41.14677 | 1.41846 | 2 |
| WB | Albesa | 41.76036 | 0.67022 | 267 | VQ | Constantí | 41.17130 | 1.16774 | 112 |
| WG | Algerri | 41.80104 | 0.64804 | 301 | XE | Tarragona | 41.10393 | 1.20100 | 5 |
| WX | Camarasa | 41.91780 | 0.88175 | 668 | C7 | Tàrrega | 41.66695 | 1.16234 | 427 |
| X6 | Artesa de Segre | 41.92173 | 1.02901 | 366 | WL | Sant Martí de Riucorb | 41.57236 | 1.08820 | 413 |
| | | | | | XX | Tornabous | 41.68835 | 1.04476 | 291 |

**Table A1.** List of Automatic Weather Stations (AWS) from the Servei Meteorològic de Catalunya (SMC) used in the study.



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
