# Peer review of "The Marinada Fall Wind in the Eastern Ebro Sub-basin: Physical Mechanisms and Role of the Sea, Orography and Irrigation"

_EGUsphere, 2024_

## Author Response (AR1)

**Answers to the reviewers' comments**

*Thanks to both reviewers for their fruitful reviews, which have led to some nice improvements to the article. In short, almost all suggestions have been incorporated into the revised version. Below is our response to the comments. Note that the reviewers' comments are in regular font and our responses are in italics.*

**RC 1 – 01/04/2024**

**General Remarks**

The numerical study consisting of high-resolution mesoscale simulations on two case study days from a field study in the Ebro sub-basin is designed well. The mesoscale simulations were conducted using the Meso-NH mesoscale model coupled with the SURFEX surface model.

Results of numerical simulations are in general in good agreement with the observations. High-resolution mesoscale simulations provided an opportunity for a detailed analysis of physical processes governing evolution of the Miranda fall winds. An exhaustive analysis presented in the manuscript elucidated the mechanisms driving circulations in Ebro sub-basin including conditions leading to onset, strengthening, and decay of Marinada winds. The analysis of atmospheric circulations in the region is sound and provides a valuable insight in the interplay between different forcings and resulting flow patterns in complex terrain.

The manuscript is organized well, the presentation and analysis of numerical results is clear, supported by well-designed figures. One weakness of the manuscript is that while it provides interesting insights about atmospheric circulations in the Catalan region it does not include any substantial new concepts, ideas, or methods within the scope of the journal.

Taking the above into account I recommend the manuscript for publication in the journal Atmospheric Physics and Chemistry after suggestions for minor revisions listed below under Specific Remarks are addressed.

*Thank you for the helpful reviews that led to nice improvements of the article. All points and suggestions have been considered and included in the revised version of the article.*
*Below is a brief description of the changes made to each item, with reference to the line in the revised version for substantial changes. Note that the reviewers' comments are in regular font and our responses are in italics.*

**Specific Remarks**

- Subsection 2.3 – The title is incomplete; it should probably be: "Weather situation of the two case study days"
  → *Modified*

- Line 136 - It is not clear what are "action centers."
  → *Term replaced by "high and low pressure centers"*

- Line 137 – Is the wind parallel to the coast thermal wind?
  → *The paragraph describing the two weather situations has been improved and the description refined (cf l. 151 – 171 of the revised version). The south-westerly flow parallel to the coast is probably a branch of the thermal low, which could be diverted by the Catalan pre-coastal range and the Pyrenees. However, it is not possible to state this with certainty with the data we have, and we prefer to limit ourselves to a factual description of the meteorological situation, without speculating too much on the origin of this branch.*

- Line 155 – Instead of "looser mesh" better would be "stretched mesh."
  → *Modified*

- Line 177 – Instead of "pedo -ransfer" it should be "pedotransfer."
  → *Modified*

- Line 247 – This is the first sentence of the Section No. 3, so it is not clear what "this" refers to. The sentence should be rewritten.
  → *Modified, the first sentence is now:*
  *"The current study uses the SMC AWS observations to compare the spatio-temporal evolution of the model's wind, temperature and humidity variables at the surface with the observations."*

- Line 252 – It is not clear how wind can blow from west to north also, previous sentence talks about sea breeze that develops during the day and now this refers to night? This is confusing.
  → *Rewritten for clarity (l.283 of revised version): The reference to the night has been changed to a reference to the change in wind direction after sunrise, typical of the sea breeze.*

- Line 252 – It is stated that "After sunrise it turns south," this would mean that wind blows from the north, I am guessing that it should be: "it turns southerly," i.e. it blows from the south.
  → *We agree with your point and the sentence has been rewritten*

- Line 264 – It should be "has decreased."
  → *Modified*

- Line 464 – It is stated that "Another topographic feature is important to consider and should be studied further as it is anthropogenic: irrigation." Also, "irrigation" is not "topographic feature,"

but a "land use feature."
> → *"Topographic feature" has been turned into "land surface feature" to be more consistent with the previous sentence.*

*In addition to the comments of the reviewers, please note that four sentences have been added l.176, 215, 466 and 585 to refer to the recently published work by Lunel et al. (2024) on irrigation breeze circulation in the same region, carried out using the same model configuration. This reference allows to support some points of the present work.*

**RC2 - 08/04/2024**

*Thank you for the fruitful feedbacks that led to nice improvements of the article. In particular, the introduction has been greatly improved following your comments. More generally, all your suggestions have been taken into account and incorporated into the revised version of the article.*
*Below is a brief description of the changes for each of your comments, with reference to the line in the revised version for substantial changes. Note that the reviewers' comments are in regular font and our responses are in italics.*

**General/mayor comments:**

This paper presents a study about the Marinada, a local wind of the northeastern Ebro basin in Catalonia. The study focused on unravelling the dynamics governing its formation and spatio-temporal characteristics, using observations and simulations, in order to improve the modeling and forecasting capabilities.

1) The introduction gives an overview of the various local winds that have been observed in different locations or are very typical for the Embo Basin. However, the introduction should be improved. Some connection between local, and yet general types of flows such as sea breeze, slope winds, and the bora wind, with local ones in the Ebro basin such as Cierzo and Marinada (which are unknown to the wider research community) is not obvious. I propose a better connection/discussion/rewriting of the presented overview of the conclusions of various studies mentioned (in the Introduction). The result of a better discussion would give a better motivation for research, which is also not clear enough now.

→ *Modified. The introduction has been reorganized and improved, and the local winds are now more detailed, including their relation to the Marinada. The improved part of the introduction is lines 66 to 98 in the revised version.*

2) Is the Marinada a local air circulation that has its own branches or is it a type of local wind. It would be better to be more precise in stating this phenomenon.

→ *Indeed, throughout the article we have used the term circulation in a broad sense, without implying that it necessarily has an observable return flow. To remove this ambiguity, the term has been replaced by "wind" in most cases throughout the text, and sometimes retained where appropriate.*

3) The methodology is well covered, but the organization of the article needs improvement(s). I suggest successive calling of figures (in order) and not jumping forward – back, e.g. from Fig. 4 to Fig. 12 (at page 11) or first Fig. 2 and only after Fig. 1.

→ *The organization has been improved. The figures are presented one after the other, with greater attention being paid to the progressive, additive contribution of the figures to the understanding of each stage of the Marinada.*

**Detailed comments in text:**

- Page 2, Lines 30-32: Please rephrase the statement… The penetration length of the sea breeze is typically about 15 to 30 km (Pokhrel and Lee, 2011; Crosman and Horel, 2010) and can reach up to 85 km as documented by Simpson et al. (1977).
  ... into…
  The penetration length of the sea breeze can be within the range of 15  km to 85 km depending on the topographic features of the coastal region and the synoptic wind if it has developed (e.g., Simpson et al., 1977; Crosman and Horel, 2010; Pokhrel and Lee, 2011; Poljak et al., 2014).
  - with new reference: Poljak et al (2014), ANGEO
  → *Modified as suggested*

- Page 2, Lines 36-37; please replace for 80 % of the sea breezes … with... for 80 % of the sea breezes events.
  → *Modified as suggested*

- Page 3, Line 69; What is the Cierzo?
  → *The description of the Cierzo has been added in line 76. In short, the Cierzo is the strong northwesterly wind that blows in the lower part of the Ebro valley.*

- Page 3, Lines 65-76; Here it would be good to refer to the figure of the area (Fig. 1?), because not everyone is familiar with the features of the Ebro  basin. Please check where Fig. 1 (from page 5) is referenced.
  → *We agree, reference added.*

- Page 4, Line 118; Please delete the extra bracket from… "Alt Camp") in …
  → *Done*

- Page 5 (hereafter); Perhaps such are the rules of the journal, but the description of the table goes above the table.
  → *The descriptions for all the tables are now placed above.*

- Page 7, subchapter 2.3; The proposal is to put the differences between these two days in a table for easier monitoring of the results.
  → *This is a nice suggestion. The weather situations have been summarized in Table 3 in the revised version.*

- Page 7, Line 149; Why is Fig. 1 mentioned only now in the text (after Fig. 2)?
  → *Fixed*

- Page 8, Line 189; Please explain what the C3 crop is.
  → *Short description added at line 218-219.*
  *"C3 crops are plants with a C3 pathway for photosynthesis, such as wheat and soybeans."*

- Page 11, Line 262-263; The term Marinade is now defined here, but it should still be put in the Introduction.
  → *Definition added in the introduction*

- Page 12, Line 292; Please correct …fig. 2 into Fig. 2.
  → *Modified*

- Page 12, Line 280; … a land breeze is generated on the coast.
  If it is about Figs. A2-A3 (after 20 hours); please explain more the indication of the land breeze on the coast (along the white line?). According to the measurements/model over the sea, and immediately along the coast, the wind is quite stationary without changing to the night (offshore) direction.
  → *Description of the white line added in Fig 4 and concerned appendices.*
  → *Sentence mentionning the land breeze removed. It is not important to mention it for the Marinada study and, as you point out, there are too few elements to say it is a land breeze.*

- Page 14, Line 307; … a medium PGF is oriented towards the southwest..; Is it correct? The red arrows in the areas marked with the 4th to 6th rectangles in Figure 5 show the direction towards the SE.
  → *That was a mistake, it's indeed towards the south-east. This has been modified accordingly.*

- Page 17: Lines 344-345 & 365; It is necessary to clarify the statement that ...the general circulation is west… (located? Or westerly wind?). How can the circulation (something circular) be west in general? I think that the term circulation is not used precisely enough through the text; vertically there is an upper and (usually opposite) lower branch of the circulation or horizontally there is the left and right side of the circulation system.
  → *Cf answer to 2nd major point: "Circulation" has been replaced by "wind" in these two cases.*

- Page 27; Line 502: Please correct, …2h00 later,
  → *replaced by "2 hours"*

- Page 30, Lines, 566-567; Is the Marinada a local air circulation that has its own branches or is it a type of local wind in some layer. It would be better to be more precise in stating this

phenomenon.

*→ Cf answer to 2nd major point: With the proposed distinction between circulation and wind, it is then a local wind.*

*In addition to the comments of the reviewers, please note that four sentences have been added l.176, 215, 466 and 585 to refer to the recently published work by Lunel et al. (2024) on irrigation breeze circulation in the same region, carried out using the same model configuration. This reference allows to support some points of the present work.*